# High-throughput phenotypic screen and transcriptional analysis identify new compounds and targets for macrophage reprogramming

Guangan Hu [1,2✉], Yang Su[1,2], Byong Ha Kang [1,3], Zhongqi Fan[1,2], Ting Dong[1,2], Douglas R. Brown[1,2], Jaime Cheah[1,2], Karl Dane Wittrup [1,3] & Jianzhu Chen [1,2✉]

Macrophages are plastic and, in response to different local stimuli, can polarize toward multi-dimensional spectrum of phenotypes, including the pro-inflammatory M1-like and the anti-inflammatory M2-like states. Using a high-throughput phenotypic screen in a library of ~4000 FDA-approved drugs, bioactive compounds and natural products, we find ~300 compounds that potently activate primary human macrophages toward either M1-like or M2-like state, of which ~30 are capable of reprogramming M1-like macrophages toward M2-like state and another ~20 for the reverse repolarization. Transcriptional analyses of macrophages treated with 34 non-redundant compounds identify both shared and unique targets and pathways through which the tested compounds modulate macrophage activation. One M1-activating compound, thiostrepton, is able to reprogram tumor-associated macrophages toward M1-like state in mice, and exhibit potent anti-tumor activity. Our compound-screening results thus help to provide a valuable resource not only for studying the macrophage biology but also for developing therapeutics through modulating macrophage activation.

[1] Koch Institute for Integrative Cancer Research, Massachusetts Institute of Technology, 77 Massachusetts Avenue, Cambridge, MA 02139, USA. [2] Department of Biology, Massachusetts Institute of Technology, 77 Massachusetts Avenue, Cambridge, MA 02139, USA. [3] Department of Biological Engineering, Massachusetts Institute of Technology, 77 Massachusetts Avenue, Cambridge, MA 02139, USA. ✉email: gahu@mit.edu; jchen@mit.edu

Macrophages are a key class of phagocytic cells that readily engulf and degrade dying/dead cells and invading bacteria and viruses. As such, macrophages play an essential role in development, tissue homeostasis and repair, and immunity[1]. Consistently, macrophages are generated during early ontogeny and throughout the adult life. In mammals, the first wave of macrophages is generated from the yolk sac and gives rise to macrophages in the central nervous system, i.e., microglia. The second wave of macrophages is generated from fetal liver and gives rise to alveolar macrophages in the lung and Kupffer cells in the liver among others. After birth, macrophages are generated from the bone marrow where hematopoietic stem cells give rise to monocytes, which differentiate into tissue resident macrophages upon migration from blood into specific tissues[2,3].

A remarkable feature of macrophages is their plasticity, the ability to respond to local stimuli to acquire different phenotypes and functions so as to respond to changing physiological needs[3]. Macrophage plasticity underlies their ability to be activated toward a spectrum of phenotypes and acquire diverse functions. One extreme is the classically activated proinflammatory M1 macrophages and the other extreme is the alternatively activated anti-inflammatory M2 macrophages[4]. By expressing inflammatory cytokines, such as IFNγ and TNFα, and reactive oxygen species, M1 macrophages mediate antimicrobial and antitumor responses, but can also cause inflammation and tissue damage if hyperactivated. In contrast, by expressing anti-inflammatory cytokines, such as IL-10, TGFβ, and arginase, M2 macrophages mediate tissue repair, but can also mediate fibrosis if dysregulated. While M1 and M2 serves to define the opposite activating states of macrophages in simplistic manner, most macrophages exhibit multidimensional spectrum of phenotypes in response to various physiological and pathological signals[2,3,5,6]. By transcriptional profiling of human monocyte-derived macrophages (hMDMs) in response to 29 different stimuli, such as pro- and anti-inflammatory cytokines, Xue et al. identified 49 gene expression modules that are associated with macrophage activation[6]. Although our understanding of macrophage biology is progressing rapidly, many aspects of macrophage activation/plasticity remain poorly defined. In particular, how small molecules modulate macrophage activation remains to be elucidated.

Because of their critical function in maintaining tissue homeostasis and repair, dysregulation of macrophage polarization has been implicated in contributing to many human diseases including cancer, fibrosis, obesity, diabetes, and infectious, cardiovascular, inflammatory, and neurodegenerative diseases[7–9]. For example, tumor-associated macrophages (TAMs) are one of the most abundant immune cells present in solid tumors. Clinical and experimental studies have shown that TAMs produce various membranous and soluble factors that enhance tumor cell growth and invasion as well as suppress antitumor immune responses to allow cancer cells to escape immune surveillance[10,11]. TAMs are derived from circulating monocytes in the tumor microenvironment, which progressively skews macrophages into the immunosuppressive state, phenotypically resembling M2-activated macrophages[12,13]. Several studies have shown that reprogramming M2-like TAMs toward M1-like macrophages is associated with expression of a strong antitumor activity[14,15]. In a remarkable synergy, cyclophosphamide-activated macrophages efficiently eliminate leukemia cells in refractory bone marrow microenvironment in combination with monoclonal antibody therapeutics[16,17]. Thus, repolarizing TAMs toward a proinflammatory, antitumorigenic M1-like state may prove an efficient approach to cancer immunotherapy either alone or in combination with antibody therapeutics. More broadly, as dysregulation of macrophage activation has emerged as a key determinant in many disease development and progression[9], modulation of macrophage activation could be a fruitful approach for disease intervention.

In this study, we develop a high-throughput phenotypic screen for small molecule compounds that can reprogram polarization of primary human macrophages. Transcriptional profiling of macrophages following treatment with selected compounds identify both shared and unique targets and pathways in macrophage polarization. M1-activating compound thiostrepton is further shown to reprogram TAMs toward M1-like macrophages in mice and exhibit potent antitumor activity either alone or in combination with a monoclonal antibody. Our results reveal a remarkable plasticity of macrophage polarization through many new pathways and provide a valuable resource not only for studying the macrophage biology but also for developing novel therapeutics or repositioning known drugs for treating diseases through macrophage reprogramming. Furthermore, the phenotypic screen can be extended to much larger compound libraries and in combination with transcriptional profiling is a powerful approach to elucidate the mechanism of action of small molecule compounds in macrophage polarization for precision disease intervention.

## Results

**Phenotypic screen of macrophage activation.** Human monocytes purified from fresh blood of four healthy donors were mixed at equal ratio and differentiated into macrophages in the presence of human M-CSF. The resulting hMDMs were stimulated with different known M1-activating stimuli, including lipopolysaccharide (LPS), IFNγ, TNFα, or IFNγ plus TNFα, or M2-activating cytokines, including IL-10, IL-4, or IL-13, for 24 h. The M1-activated hMDMs were round with punctate F-actin staining, whereas M2-activated hMDMs were elongated with filamentous F-actin staining (Fig. 1a and Supplementary Fig. 1a). Expression of known M1 markers including CD80 and CD86 were upregulated by IFNγ and suppressed by IL-4, while M2 markers CD206 and CD163 were upregulated by IL-4 and suppressed by IFNγ (Supplementary Fig. 1b). The Z-score for each stimulus was calculated to index its activation ability from the distributions of cell shapes between treated wells and untreated wells by T-test of an average of 1000 cells per well. The M1-activated hMDMs had an average of Z-score of −4, whereas the M2-activated hMDMs had an average of Z-score of 6 (Fig. 1b). These results are consistent with previous reports showing that M1- and M2-like human and mouse macrophages have distinct morphologies[18–21].

Based on the correlation between cell shape and macrophage activation, we developed a high-throughput screen for compounds that activate hMDMs to either M1- or M2-like state (Fig. 1c). Fresh hMDMs derived from four healthy donors at equal ratio were seeded into 384-well plates and cultured overnight in the presence of M-CSF to maintain macrophages at mostly a nonactivated stage. Macrophages in each well were then treated with one of 4126 compounds, including 2086 bioactive compounds, 760 FDA-approved drugs, and 1280 natural products (Fig. 1d), at a final concentration of 20 μM for 24 h. Cell images were taken by high-content scanning microscope and cell shapes were quantified by CellProfiler (Fig. 1e)[22]. Based on Z-score cutoffs: −4 for M1-activated macrophages and 6 for M2-activated macrophages, 127 and 180 compounds were identified, respectively, to activate human macrophages toward M1-like state (referred to as M1-activating compounds) and M2-like state (referred to as M2-activating compounds) (Fig. 1f and Supplementary Data 1). Overall, 98 of 127 (77%) M1-activating and 166 of 180 (92%) M2-activating compounds are FDA-approved drugs (Fig. 1g). Text mining identified 119 known

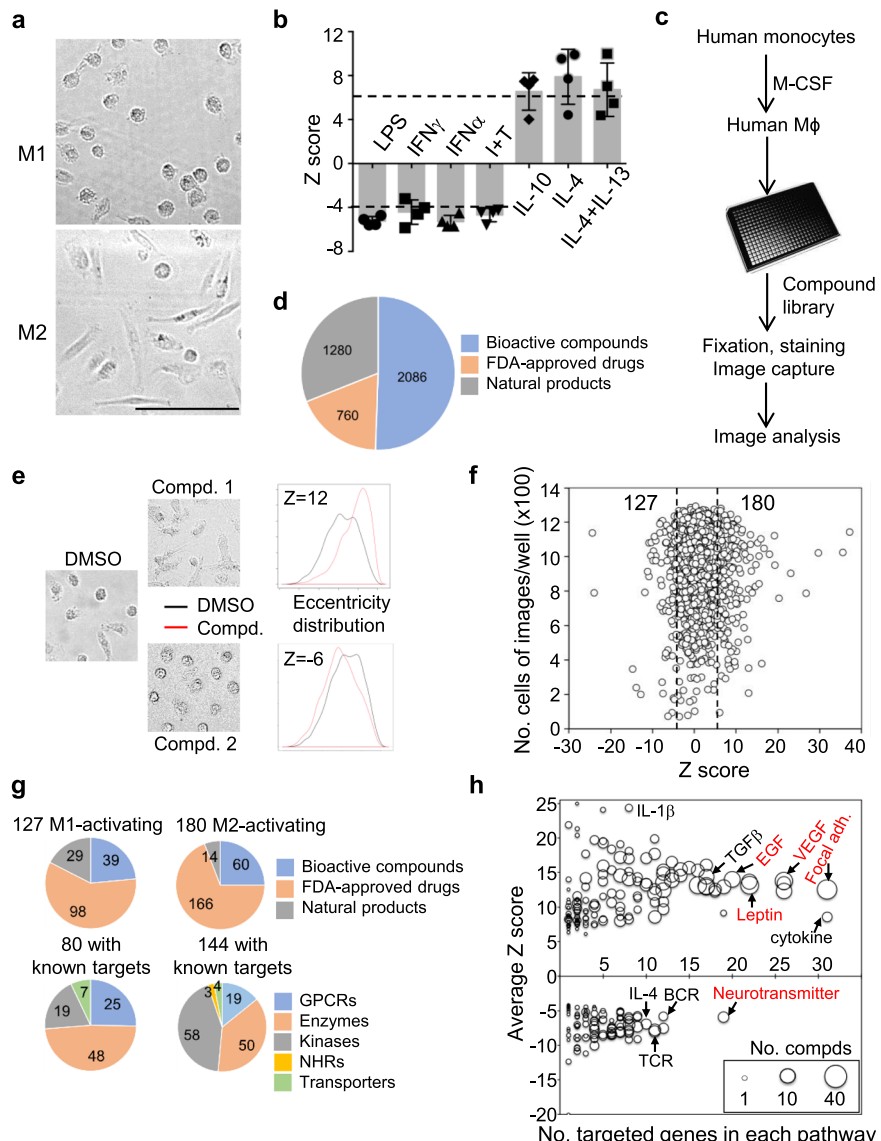

**Fig. 1 A high-throughput screen for compounds that activate human macrophages. a**, **b** hMDMs were cultured for 24 h in the presence of LPS, IFNγ, TNFα, IFNγ plus TNFα (I+T), IL-10, IL-4, or IL-13. Shown are examples of cell morphologies of M1-activated macrophages by IFNγ and M2-activated macrophages by IL-4 (**a**) from three independent experiments and calculated Z-scores for each stimulus (**b**) from four independent experiments. Scale bar: 100 μm. Data are presented as mean ± sd. The Z-score was calculated by T-test to measure the difference of cell morphology between treatment and control. Stimuli had negative Z-scores when induced cells to round morphology and positive scores when induced cells to elongated morphology. **c** The flowchart of high-throughput screen and data analysis. Equally mixed human monocytes isolated from fresh blood of four healthy donors were cultured in vitro with 50 ng/mL M-CSF for 7 days. hMDMs were trypsinized and plated on 384-well plates (5000 cells/well in 50 μL). Cells were recovered in 10 ng/mL M-CSF for 16 h and then treated with compounds for 24 h. Cells were washed, fixed, and stained with phalloidin and DAPI. The plates were scanned with a high-content microscope with six fields per well to quantify the cell number and cell morphology. **d** Composition of compound libraries used in the screen. **e** Examples of cell shape changes induced by two compounds and their corresponding Z-scores as compared to DMSO controls. The cell eccentricity was calculated to measure the cell morphology. The Z-score was calculated by T-test to measure the difference in cell morphologies between each compound and DMSO control. **f** Plot of Z-scores of 4126 compounds and number of cells captured in each well. The dash lines are the cutoffs for M1 activation (left) and M2 activation (right) based on the average of Z-scores from **b**. **g** Classification of identified compounds based on their origin and function of their known targets. **h** Pathway analysis of known targets of identified M1- or M2-activating compounds. Each dot is one specific pathway having protein targets by compounds and dot size refer to the number of compounds. The average Z-score (y-axis) and number of compounds that have protein targets belongs to one specific pathway are plotted. Selected known (black) and new (red) pathways associated with macrophage activation are indicated.

protein targets for 80 of the 127 M1-activating compounds and 220 protein targets for 144 of the 180 M2-activating compounds. The targets include G-protein coupled receptors (GPCRs), enzymes, kinases, nuclear hormone receptors (NHRs), and transporters (Fig. 1g). Many targets of M1- and M2-activating compounds belong to the families of histone deacetylases and

vascular endothelial growth factor (VEGF) receptors, respectively (Supplementary Fig. 2). Some known regulators of macrophage polarization, such as STAT3, FYN, MAP2K1, and CDKs[6], were rediscovered. Pathways analysis of the protein targets identified known pathways, such as IL-4, IL-1β, and TGFβ pathways, and new pathways, such as neurotransmitter, leptin, epidermal

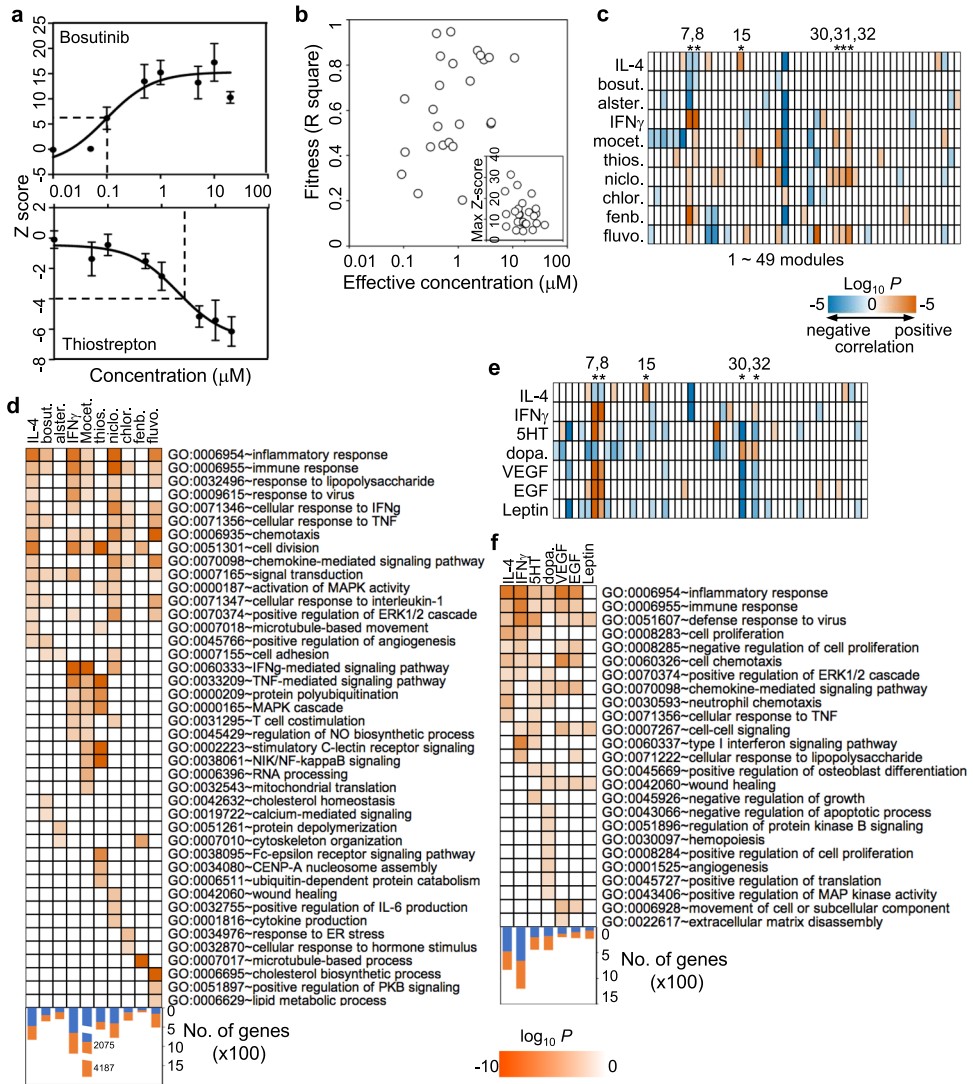

**Fig. 2 Validation of macrophage activation induced by compounds or by ligands of the identified new pathways. a, b** Morphology changes induced by selected compounds are dosage dependent. Dosage responses were calculated based on the measurement of $Z$-scores at different concentrations of the compound in a Michaelis–Menten model. Shown are dosage response curves of M1-activating (thiostrepton, $n = 6$) and M2-activating (bosutinib, $n = 4$) compounds (**a**). Data are presented as mean ± sem. Overall, 25 of the 30 tested compounds had typical dosage-dependent response (**b**). Effective concentration (EC) was defined as the concentration of compounds inducing cell morphology changes to reach the cutoffs of either M1 or M2. EC, fitness ($R^2$), and max $Z$-score were calculated by the Michaelis–Menten equation. Data were summarized from three independent experiments. **c** GSEA of transcriptional responses to eight selected compounds and controls (IL-4 and IFNγ). Duplicate hMDM samples were treated with two M2-activating and six M1-activating compounds as well as IL-4 and IFNγ for 24 h. Gene expression levels were measured by RNA-seq separately. GSEA preranked analysis was performed based on the whole genome gene list ranked on gene expression changes using a gene set of 49 transcriptional modules in response to 29 stimuli in hMDMs[6]. bosut.: bosutinib, alster.: alsterpaullone, mocet.: mocetinostat; thios.: thiostrepton, niclo.: niclosamide, chlor.: chlorhexidine, fenb.: fenbendazole, fluvo.: fluvoxamine. The numerical numbers at the top indicate module numbers identified by Xu et al.[6]. **d** GO enrichment analysis of DEGs induced by each compound and positive controls. The numbers of DEGs that are upregulated (orange) and downregulated (blue) are indicated. **e** GSEA of transcriptional responses to six ligands of the identified new pathways in Fig. 1h. dopa.: dopamine, 5HT: serotonin. Duplicate hMDM samples were stimulated with each ligand and analyzed by RNA-seq separately. **f** GO enrichment analysis of DEGs induced by the ligands and positive controls. The numbers of DEGs that are upregulated (orange) and downregulated (blue) are indicated.

growth factor (EGF), and VEGF signaling pathways, in macrophage activation (Fig. 1h and Supplementary Data 2).

**Validation of selected compounds and pathways.** To validate the effect of identified compounds on macrophage activation, we assayed dosage responses of the commercially available top list of compounds to determine their effective concentration (EC) on cell shape change. Overall, 20 of 23 selected M1-activating and 4 of 6 M2-activating compounds showed strong dosage effects with

an EC below 10 μM (Fig. 2a, b and Supplementary Table 1). To determine whether the compounds activate macrophages at the transcriptional level, we performed RNA-seq analyses of hMDMs following treatment with six M1-activating (mocetinostat, thiostrepton, niclosamide, chlorhexidine, fenbendazole, and fluvoxamine) and two M2-activating (bosutinib and alsterpaullone) compounds at the EC for 24 h. These compounds induced diverse transcriptional responses with variable number of differentially expressed genes (DEGs) to similar degrees as those induced by

IL-4 or IFNγ (Supplementary Fig. 3a and Supplementary Data 3). To explore the functional differences of hMDMs induced by compounds, gene set enrichment analysis (GSEA) was performed against the previously identified 49 gene expression modules in response to 29 different stimuli in hMDMs[6]. Similar to IFNγ, the six M1-activating compounds upregulated the gene expression of typical M1 modules (#7, #8) induced by IFNγ, as well as chronic inflammation TPP modules (#30, #32) induced by TNFα/PGE2/P3C (Fig. 2c). The M1-activating compounds also downregulated the modules (#26, #27) similarly as LPS. The two M2-activating compounds downregulated the gene expression of typical M1 modules although they did not upregulate the gene expression modules (module #15) induced by IL-4 (Fig. 2c). Consistently, all M1-activating compound upregulated expression of the classical M1 markers CD80 and CD86 and downregulated expression of the classical M2 markers CD163 and CD206 (Supplementary Fig. 3c). Both M2-activating compounds downregulated M1 markers. Moreover, based on function enrichment analysis of the DEGs, all eight compounds induced consensus pathways related to inflammatory response, chemotaxis/chemokine-mediated signaling and response to IFNγ and TNFα (Fig. 2d). These results suggest that select compounds modulate macrophage activation at the transcriptional levels.

We also analyzed transcriptional responses of hMDMs to ligands of new pathways, including serotonin (5HT), dopamine, VEGF, EGF, and leptin by RNA-seq. Each ligand induced diverse transcriptional responses (Supplementary Fig. 3b and Supplementary Data 4). In particular, 5HT, VEGF, EGF, and leptin upregulated gene expression of the typical M1 modules (#7, #8) but downregulated gene expression of the TPP modules (#30, #32) (Fig. 2e). In contrast, dopamine downregulated gene expression of the typical M1 modules but upregulated the TPP modules (Fig. 2e), suggesting these ligands regulate different aspects of macrophage activation. Function enrichment analysis of the DEGs identified induction of pathways related to inflammatory response, chemotaxis/chemokine-mediated signaling, and wound healing by these ligands (Fig. 2f). Taken together, these results suggest that the compounds as well as upstream signals of their protein targets modulate macrophage activation.

**Reprogramming screen of compounds on polarized macrophages.** To investigate whether the identified compounds could reprogram or reactivate macrophages after M1- or M2-like differentiation, we rescreened the hits (as identified in Fig. 1f) on M1- or M2-activated macrophages. hMDMs were differentiated into M2-like macrophages by IL-4 plus IL-13 or M1-like macrophages by IFNγ plus TNFα. After removing the differentiating cytokines, M2-like macrophages were treated with each of the 166 M1-activating compounds and M1-like macrophages were treated with each of the 180 M2-activating compounds at a final concentration of either 5 or 10 μM. Twenty-four hours later, cell images were taken and cell shapes were quantified. Based on the same Z-score cutoff, 37 M1-activating and 21 M2-activating compounds were identified to induce cell shape changes at the concentration of both 5 and 10 μM (Fig. 3a, b). Dosage responses were carried out with 40 commercially available compounds (21 M1-activating and 19 M2-activating) on polarized macrophages. Overall, 17 of the M1-activating (81%) and 18 of the M2-activating (95%) compounds had typical dosage-dependent response with an EC below 10 μM, and induced statistical significant changes of cell shape (Fig. 3c and Supplementary Table 2).

We also rescreened the hits on differentiated macrophages in the presence of differentiating cytokines. Surprisingly, more compounds exhibited significant effects on cell shape changes in the presence of these cytokines (67 M1-activating and 55 M2-activating) than in absence of these cytokines (46 M1-activating and 25 M2-activating) at the same compound concentration of 5 μM (Fig. 3d, e). Consistently, 28 of the 37 M1-activating and 18 of the 21 M2-activating compounds were identified again to induce significant cell shape change at the concentration of 5 μM. In the dosage response assay, the ECs of many M1-activating compounds were lower in the presence of cytokines than in the absence of cytokines (Supplementary Fig. 4), suggesting that the presence of differentiating cytokines makes macrophages more sensitive to reprogramming. Together, these data show that many identified compounds are capable of reprograming already polarized macrophages.

**Shared and unique effects of identified compounds on macrophage transcription.** To broadly validate the identified compounds on macrophage activation (reprogramming) and to shed light on how the compounds activate macrophages, we selected 17 M1-activating and 17 M2-activating compounds with ECs below 5 μM and performed transcriptional profiling by RNA-seq. M2-like macrophages induced by IL-4 plus IL-13 were treated with each of the 17 M1-activating compounds at its ECs for 24 h. Similarly, M1-like macrophages induced by IFNγ plus TNFα were treated with each of the 17 M2-activating compounds at its ECs for 24 h. Different compounds upregulated and downregulated different number of genes (Fig. 4a), and a total of 7344 genes exhibited at least a twofold change after exposure to at least one compound (Supplementary Data 5). Hierarchical clustering of Pearson's correlations of DEGs induced by compounds as well as by IFNγ and IL-4 showed that all 17 M1-activating compounds clustered together with IFNγ and all 17 M2-activating compounds clustered together with IL-4 (Fig. 4b). Principal component analysis of global transcriptional response showed that M1-like macrophages, M2-like macrophages treated with IFNγ, M1-like macrophages treated with IL-4, and M1-like macrophages treated with M2-activating compounds grouped together, whereas M2-like macrophages and M2-like macrophages treated with M1-activating compounds grouped together (Supplementary Fig. 5a). Although most compounds as well as IL-4 moderately modulated the global gene expression, GSEA showed that all M1-activating compounds clustered together and upregulated typical M1 modules (#7, #8) and the TPP modules (#30, #32) (Fig. 4c). All M2-activating compounds clustered together and downregulated the typical M1 modules (#7, #8) and the TPP modules (#30, #32). The modules (#26, #27), which are downregulated by LPS, were also downregulated by M1-activating compounds but upregulated by M2-activating compounds. Moreover, expression of the typical M1 markers CD80 and CD86 was upregulated by M1-activating compounds and suppressed by M2-activating compounds, while expression of the M2 markers CD206 and CD163 was upregulated by M2-activating compounds and suppressed by M1-activating compounds (Supplementary Fig. 5c). These results were further validated at transcriptional level by qPCR and at protein level by flow cytometry (Supplementary Fig. 5d, e).

To investigate the common denominators of macrophage activation, a reverse engineering regulatory network was assembled by ARACNe[23] based on mutual information between each gene pair computed from the compound-perturbing expression profiles. Top 10% central hub genes inferred from the network ($n = 1255$ most interconnected genes) collectively participated in 98,048 interactions. Many of the top central hub genes or regulators, such as GBP1[24], FAM26F[25], and STAT1[26], have been shown to play essential roles in macrophage activation and function (Supplementary Fig. 6). We performed GO enrichment analysis of these hub genes with visualization of

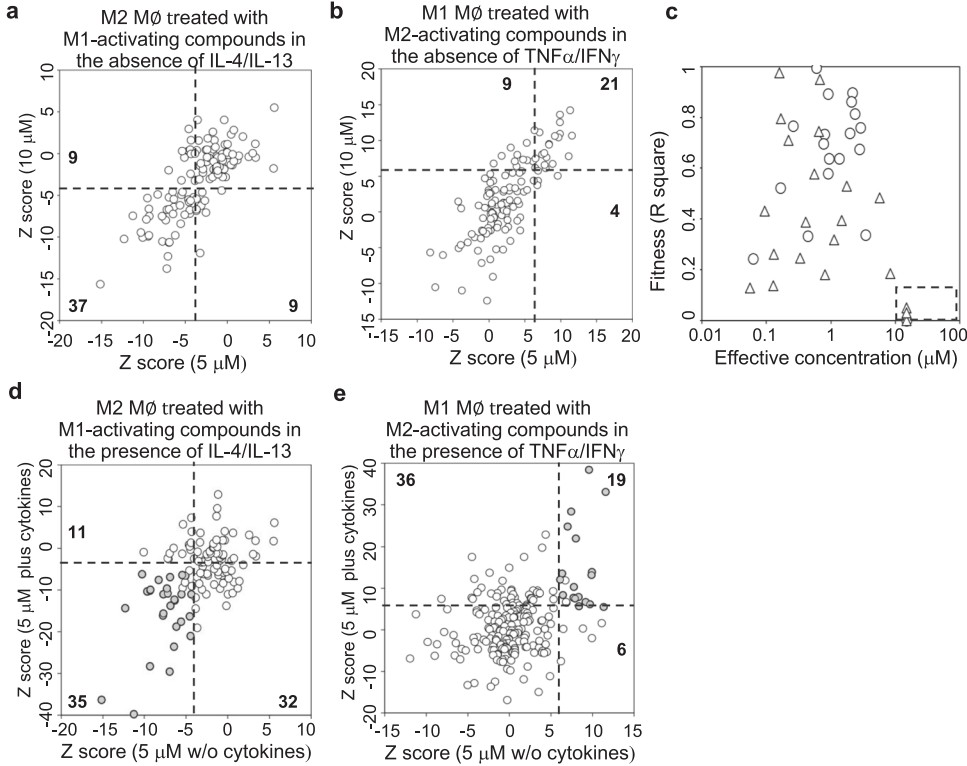

**Fig. 3 Reprogramming screen of compounds on differentiated macrophages. a** hMDMs were differentiated into M2 by IL-4 plus IL-13 and then treated with each of the 127 identified M1-activating compounds at either 5 or 10 μM for 24 h in the absence of differentiating cytokines. Shown are comparisons of Z-scores between 5 and 10 μM of compounds. **b** hMDMs were differentiated into M1 by IFNγ plus TNFα and then treated with each of the 180 identified M2-activating compounds at either 5 or 10 μM for 24 h in the absence of differentiating cytokines. Shown are comparison of Z-scores between 5 and 10 μM of compounds. **c** The effective concentration of 40 selected M1- or M2-activating compounds calculated from the dosage assays. EC and fitness of 21 M1-activating (triangle) and 19 M2-activating compounds (circle) were calculated by the Michaelis–Menten equation and plotted. Data were summarized from three independent experiments. **d–e** hMDMs were differentiated into either M2 by IL-4 plus IL-13 or M1 by IFNγ plus TNFα and then treated with 127 M1-activating (**d**) or 180 M2-activating (**e**) compounds for 24 h in the presence of differentiating cytokines. Filled dots show identification of the same 37 M1-activating (**a**) and 21 M2-activating (**b**) compounds.

GO enrichment networks by BiNGO[27]. This GO-term network identified functional clusters associated with macrophage activation, including not only previously identified clusters[6] of immune response, leukocyte or lymphocyte activation, and catabolic and metabolic process, but also new clusters of stress response, cell migration, protein transport, secretion, cell proliferation, ion homeostasis, phosphorylation and signaling, Ras signal transduction, as well as tissue remodeling and wound healing (Fig. 4d and Supplementary Data 6). Moreover, function enrichment analysis of DEGs showed that different compounds not only modulated gene expression in the common immune response pathways and chemotaxis/chemokine-mediated signaling pathway but perturbed specific (unique) pathways (Fig. 4e, f and Supplementary Fig. 5b). Consistently, these unique pathways perturbed by compounds were primarily through their putative targets. For example, M1-activating compound MS275 inhibits HDACs (histone deacetylase), which perturbed the pathway of chromatin assembly. M2-activating compound bisantrene inhibits TOP2A (topoisomerase II), which perturbed the pathway of DNA topological change (Fig. 4f), and bosutinib inhibited phosphorylation of SRC while increased the phosphorylation salt-induced kinases (Supplementary Fig. 5f) as a multiple kinase inhibitor. These data suggest that the identified compounds reprogram the differentiated macrophages through modulating the expression of genes associated with macrophage activation as well as specific pathways unique to each compound.

**Induction of macrophages to proinflammatory state by thiostrepton.** To determine if the identified compounds activate macrophages in disease setting in vivo, we selected thiostrepton, a natural cyclic oligopeptide and an approved veterinary antibiotic for treating skin infection, and tested it to activate macrophages to M1-like state. Similar to other thiopeptide antibiotics, thiostrepton inhibits the ribosome function of bacterial protein synthesis[28]. Recently, thiostrepton was shown to exhibit antiproliferative activity in human cancer cells through inhibiting proteasome and/or FOXM1 transcription factor[29–31]. Following treatment of hMDMs with 2.5 μM thiostrepton for 24 h, hMDMs were polarized to express proinflammatory cytokines TNFα and IL-1β and downregulate the M2 chemokine CCL24 (Fig. 5a). Functional enrichment analysis of the DEGs showed that IFN/NFκB pathway, TNF-mediated pathway, oxidative-reduction process, protein polyubiquitination, and cellular response to LPS were upregulated, while DNA replication, cell cycle, and cell matrix adhesion were downregulated (Fig. 5b). GSEA showed pathways of TNFα signaling via NFκB and ROS were upregulated, while pathways of E2F target and mitotic spindle were downregulated (Fig. 5c). These results show that thiostrepton regulates the expression of genes associated with proteasome and DNA replication in hMDMs, consistent with previous studies in other cell types[30–32].

To determine the effect of thiostrepton on TAM in vitro, mouse bone marrow macrophages (BMMs) were cultured in the

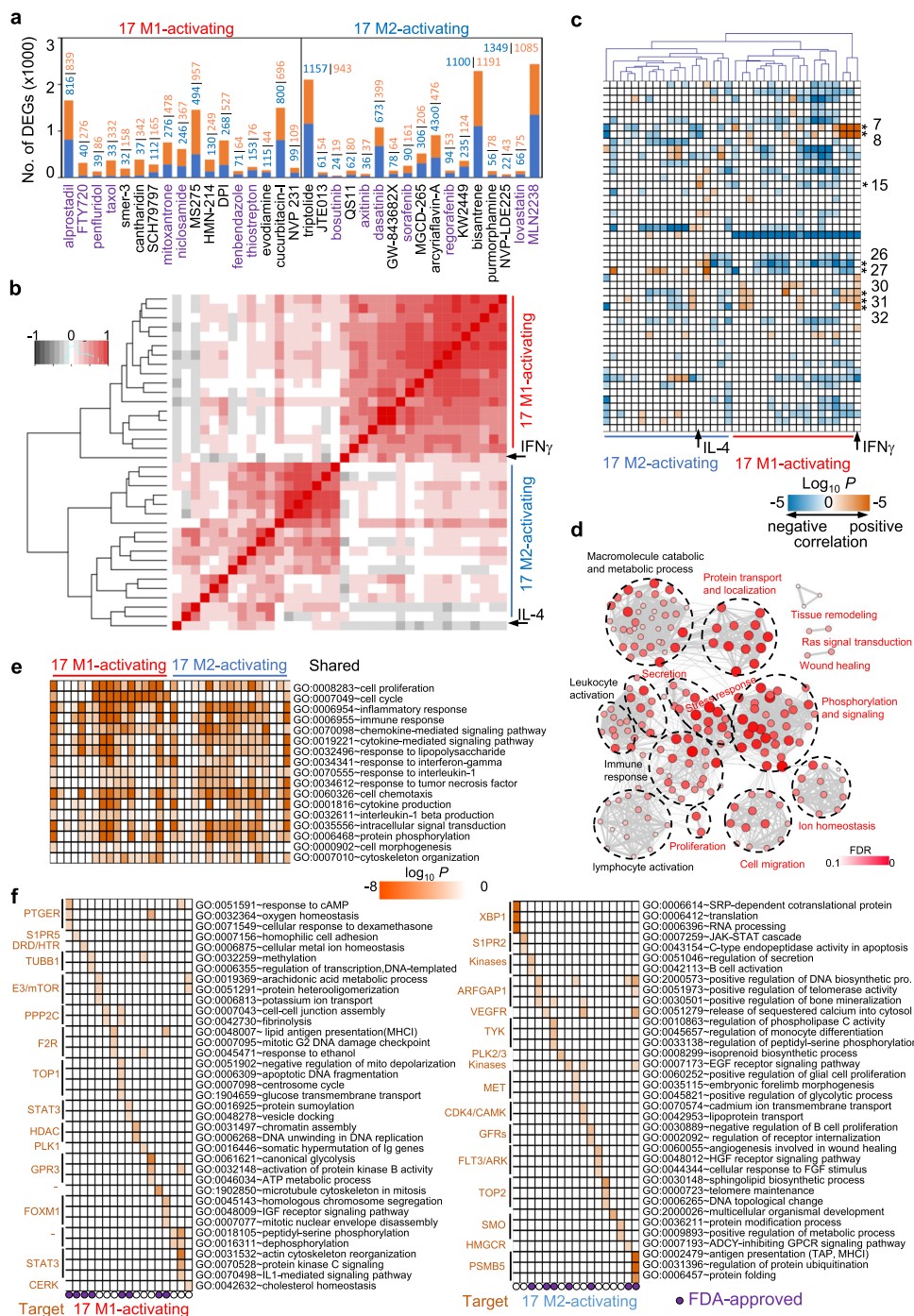

**Fig. 4 Reprogramming of differentiated macrophages by selected compounds. a** Number of DEGs induced by each compound. Orange: upregulated genes. Blue: downregulated genes. Compounds labeled in purple are FDA-approved drugs. hMDMs were differentiated into either M2 by IL-4 plus IL-13 or M1 by IFNγ plus TNFα and duplicate samples were then treated with either M1-activating or M2-activating compounds, respectively, at the effective concentrations. Controls include two differentiated M1 and M2 macrophages, M2 macrophages treated with IFNγ and M1 macrophages treated with IL-4. Gene expression in each sample was measured by RNA-seq separately. **b** Hierarchical clustering heatmap of Pearson correlation coefficients for 7620 DEGs induced by compounds as well as IFNγ and IL-4. **c** GSEA analysis of transcriptional responses to each compound as compared to IFNγ and IL-4. The numerical numbers on the right indicate module numbers identified by Xu et al.[6]. **d** Network of GO enriched terms using BiNGO on top 10% central hubs genes (*n* = 1255) of macrophage activation network. Node color and size represent the FDR values of enriched GO terms. New pathways identified in this study are labeled in red. **e**, **f** Functional enrichment analysis of DEGs induced by each compound. Shared (**e**) and unique pathways (**f**) are shown. Compound targets and FDA-approval information are indicated. The order of M1-activating and M2-activarting compounds in **e** and **f** are the same as in **a**.

conditioned medium (CM) of B16F10 tumor cells in the absence or presence of thiostrepton for 24 h. Alternatively, BMMs were cultured in the CM for 24 h first and then treated with thiostrepton for another 24 h. The expression of selected genes associated with macrophage polarization was assayed by qPCR. Thiostrepton inhibited the expression of TAM/M2-associated genes *Arg1*, *Fizz1*, *Vegfa*, *Ym1*, and *Tgfb* but upregulated the expression of M1-associated genes *Tnf*, *Il1b*, *Cxcl2*, and *Nos2*

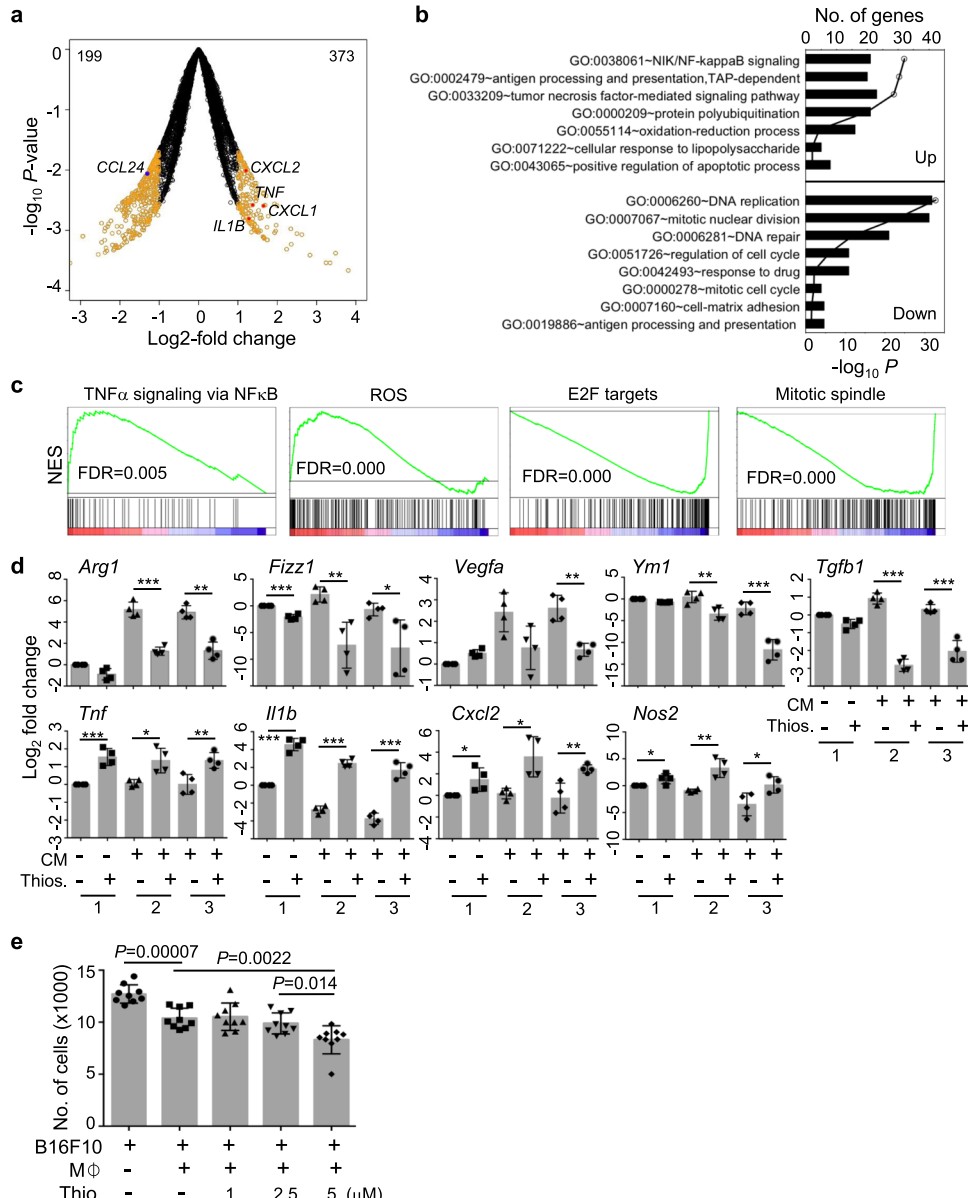

**Fig. 5 Thiostrepton induces macrophages into proinflammatory state and enhances antitumor activity in vitro. a** Volcano plot showing changes in transcription in hMDMs induced by thiostrepton ($n = 2$). hMDMs were treated with 2.5 μM thiostrepton for 24 h followed by RNA-seq. DEGs (orange) were identified by edgeR[56] at $P < 0.05$ with at least two fold-change. Data for genes that were not classified as differentially expressed are plotted in black. Filled dots show upregulated (red) and downregulated (blue) genes. **b** GO enrichment analysis of DEGs induced by thiostrepton. **c** GSEA of transcriptional response to thiostrepton. **d** Thiostrepton inhibits the development and function of TAMs in vitro. Mouse BMMs were cultured in normal medium with or without 2.5 μM thiostrepton for 24 h (group 1), or cultured in B16F10 tumor cell conditioned medium (CM) with or without 2.5 μM thiostrepton for 24 h (group 2), or cultured with B16F10 tumor cell CM for 24 h first and then treated with 2.5 μM thiostrepton for another 24 h (group 3). The transcript levels of the indicated genes were quantified by qPCR. Data are presented as mean ± sd from four independent samples per group in two independent experiments. *$P < 0.05$, **$P < 0.01$ and ***$P < 0.001$ by two-sided T-test between untreated and treated groups are indicated. **e** Thiostrepton enhances antitumor activities of macrophages. Mouse BMMs were cultured with or without thiostrepton for 24 h, and then cocultured with equal number of B16F10 melanoma cells for 12 h. The number of tumor cells was quantified by flow cytometry after subtracting macrophages from total number of cells. Data are presented as mean ± sd from three independent experiments. $P$ values by two-sided T-test are indicated.

(Fig. 5d). The effect of thiostrepton was observed whether thiostrepton was added into the CM culture or BMMs were differentiated into TAM first (compare groups 2 and 3 in Fig. 5d). Consistently, flow cytometry analysis revealed upregulation of MHCII, CD80, and iNOS but downregulation of ARG1 (Supplementary Fig. 7a). Similarly, we examined the effect of thiostrepton on IL-4/IL-13 and lactic acid-polarized BMMs. As shown in Supplementary Fig. 7b, thiostrepton inhibited the expression of *Arg1*, *Fizz1*, *Ym1*, and *Tgfb* but elevated expression

of *Tnf*, *Il1b*, *Cxcl2*, and *Ccl5* whether thiostrepton was added together with cytokines or lactic acid or after BMM polarization.

To examine whether thiostrepton-activated macrophages or CM have effects on tumor cell growth, BMMs were treated with thiostrepton for 24 h. Equal numbers of primed BMMs and melanoma cells (B16F10) were cocultured for 12 h. Significantly more melanoma cells were lost in the presence of thiostrepton-treated macrophages as compared to the untreated macrophages in a dose-dependent manner (Fig. 5e). Similarly, more

melanoma cells were lost in the CM from thiostrepton-treated macrophages than CM from untreated macrophages or heat-inactivated thiostrepton-treated CM (Supplementary Fig. 8a). To determine whether thiostreption-activated macrophages exhibit enhanced antibody-dependent cell-mediated phagocytosis (ADCP), thiostreption-activated macrophages were cocultured with equal number of human B lymphoma cells (GMB) labeled with eFluro670 dye and anti-CD20 for 2 h. Thiostrepton elevated ADCP of both human and mouse macrophages (Supplementary Fig. 8b, c). These data show that thiostrepton activates and reprograms macrophages toward a proinflammatory state and enhances their tumor-killing activity in vitro.

**Reprogramming TAMs for enhanced antitumor activity in vivo by thiostrepton.** Next, we examined whether thiostrepton has antitumor effect in vivo through activating macrophages. B16F10 melanoma cells were injected subcutaneously into syngeneic C57BL/6 (B6) mice. Six and twelve days later, tumor-bearing mice were treated with either vehicle (DMSO), melanoma specific antibody TA99[33], thiostrepton or combination of TA99 and thiostrepton by intraperitoneal (I.P.) injection. In a dosage-dependent manner (150 or 300 mg/kg), thiostrepton strongly suppressed the tumor growth alone and additively with TA99 (Fig. 6a). Since thiostrepton inhibits cell proliferation and is an antibiotic, to exclude its systematic effects on immune cells and on gut microbiome, tumor-bearing mice were treated by paratumor subcutaneous (S.C.) injection with a lower dose of thiostrepton (20 mg/kg). This local treatment also suppressed the tumor growth and exhibited additive effects with TA99 (Fig. 6b). Flow cytometry analysis of single-cell suspensions of dissected tumors at day 18 post tumor engraftment showed elevated levels of macrophages and monocytes in mice given thiostrepton or thiostrepton plus TA99 as compared to mice given vehicle or TA99 (Fig. 6c, d). Consistently, more abundant macrophages were stained positive for F4/80 by immunochemistry in tumor sections from mice treated with thiostrepton or thiostrepton plus TA99 than mice treated with vehicle or T99 (Fig. 6e). In nontumor-bearing mice, I.P. administration of thiostrepton led to increased numbers of macrophages in the spleen and bone marrow, while S. C. administration did not have significant effects on macrophage numbers (Supplementary Fig. 9a, b). In both dosing strategies, thiostrepton did not change the total bacterial counts in the gut (Supplementary Fig. 9c). Moreover, flow cytometry analysis of TAM revealed elevated levels of iNOS and CD86 and decreased levels of Arg1 in mice given thiostrepton or thiostrepton plus TA99 as compared to mice given vehicle or TA99 (Supplementary Fig. 10a–c). Interestingly, an increased number of TNFα+ IFNγ+ NK cells (but not CD8+ T cells) was found in tumors in mice given thiostrepton or thiostrepton plus TA99 as compared to mice given vehicle or TA99 (Supplementary Fig. 10d, e).

To investigate whether tumor-infiltrating macrophages were reprogrammed, we purified TAMs from B16F10 melanoma tumors from mice dosed with thiostrepton or vehicle by I.P. or S.C. at day 18 post tumor engraftment and performed RNA-seq. GSEA and functional enrichment analysis showed that thiostrepton upregulated the expression of genes associated with inflammatory response and ROS and downregulated the expression of genes associated with mitotic division in TAMs from mice treated with thiostrepton by both I.P. and S.C. (Supplementary Fig. 11 and Supplementary Data 7). The expression of the proinflammatory cytokines, including *Tnf*, *Il1b*, *Cxcl1*, and *Cxcl2*, were also significantly upregulated (Fig. 6f), consistent with the results from thiostrepton treatment of hMDMs in vitro (Fig. 5a).

To further confirm the antitumor effects of thiostrepton in vivo, we injected i.v. luciferase-expressing human B lymphoma

cells into NSG mice. Tumor-bearing mice were treated with rituximab (anti-CD20), thiostrepton, or both at 2 and 3 weeks post tumor engraftment. Quantification of tumor burden by bioluminescence imaging showed that thiostrepton alone or together with rituximab significantly reduced the tumor burden in the bone marrow (Supplementary Fig. 12a, b). Consistently, higher percentages of F4/80+CD11b+ macrophages with higher expression of MHCII were found in the bone marrow of mice treated with thiostrepton than mice given vehicle or rituximab (Supplementary Fig. 12c–f). Moreover, other M1-activating compounds, cucurbitacin I and mocetinostat, also inhibited B16F10 growth by activating macrophages both in vitro and in vivo (Supplementary Fig. 13). Taken together, M1-activating compounds could reprogram TAMs into proinflammatory macrophages to inhibit tumor growth in vivo.

## Discussion

Our high-throughput phenotypic screen is based on macrophage cell shape changes in response to compound treatment. Cell shape change is a valid phenotypic profiling of macrophage activation based on the following considerations. First, cell shape changes are mediated by changes in cytoskeleton dynamics and are known to associate with different states of cell function in general[34,35]. More specifically, both mouse and human macrophages are known to exhibit dramatically different cell shapes following activation into different phenotypes in vitro: an elongated shape for M2-like macrophages and round shape for M1-like macrophages[18,21]. Consistently, we show that known M1-activating stimuli LPS, IFNγ, and TNFα induce round cell shape, whereas known M2-activating stimuli IL-4, IL-13, and IL-10 induce elongated cell shape (Fig. 1). Similarly, GM-CSF-induced round human macrophages and M-CSF-induced elongated human macrophages exhibit M1-like and M2-like phenotypes[36], respectively, based on cytokine profiles[18,37] and the genome-wide gene expression[19,20]. Second, inducing cytoskeleton changes by extracellular stress[21] or drug paclitaxel[38] are known to lead to macrophage polarization. In our study, we also identified several compounds/drugs that modulate macrophage morphology by directly regulating actin-cytoskeleton, including paclitaxel as well as other M1-activating compounds: cytochalasin-B, fenbendazole, parbendazole, and methiazole, and M2-activating compounds: podofilox, colchicine, and vinblastine sulfate. Analyses of human macrophage responses to fenbendazole and paclitaxel further confirm that both drugs activate macrophages toward M1-like phenotype at the transcriptional and translational levels (Figs. 2c and 4 and Supplementary Figs. 3c and 5). Third, although we use cell shape change as a high-throughput readout in the initial phenotypic screen, we confirm the effects of over 40 selected compounds on macrophage activation by RNA-seq (Figs. 2 and 4) and by flow cytometry (Supplementary Fig. 5). As expected, pathway analysis of DEGs identified cell morphogenesis and cytoskeleton organization as major GO terms that are regulated by the compounds (Fig. 4c). Thus, the cell shape-based phenotypic screen with primary human macrophages is a valid approach and can be extended to much larger compound libraries as the microscopy-based cell shape profiling can be easily scaled up. As further discussed below, the combination of the phenotypic screen and transcriptional analysis could be a powerful approach to identify compounds and their mechanisms of action in macrophage activation for new drug development.

Our study identifies compounds, targets, and pathways that mediate macrophage activation and sheds light on the underlying molecular mechanisms. In our library, many compounds have known protein targets. Based on functional pathway enrichment analysis of protein targets of M1- or M2-activating compounds,

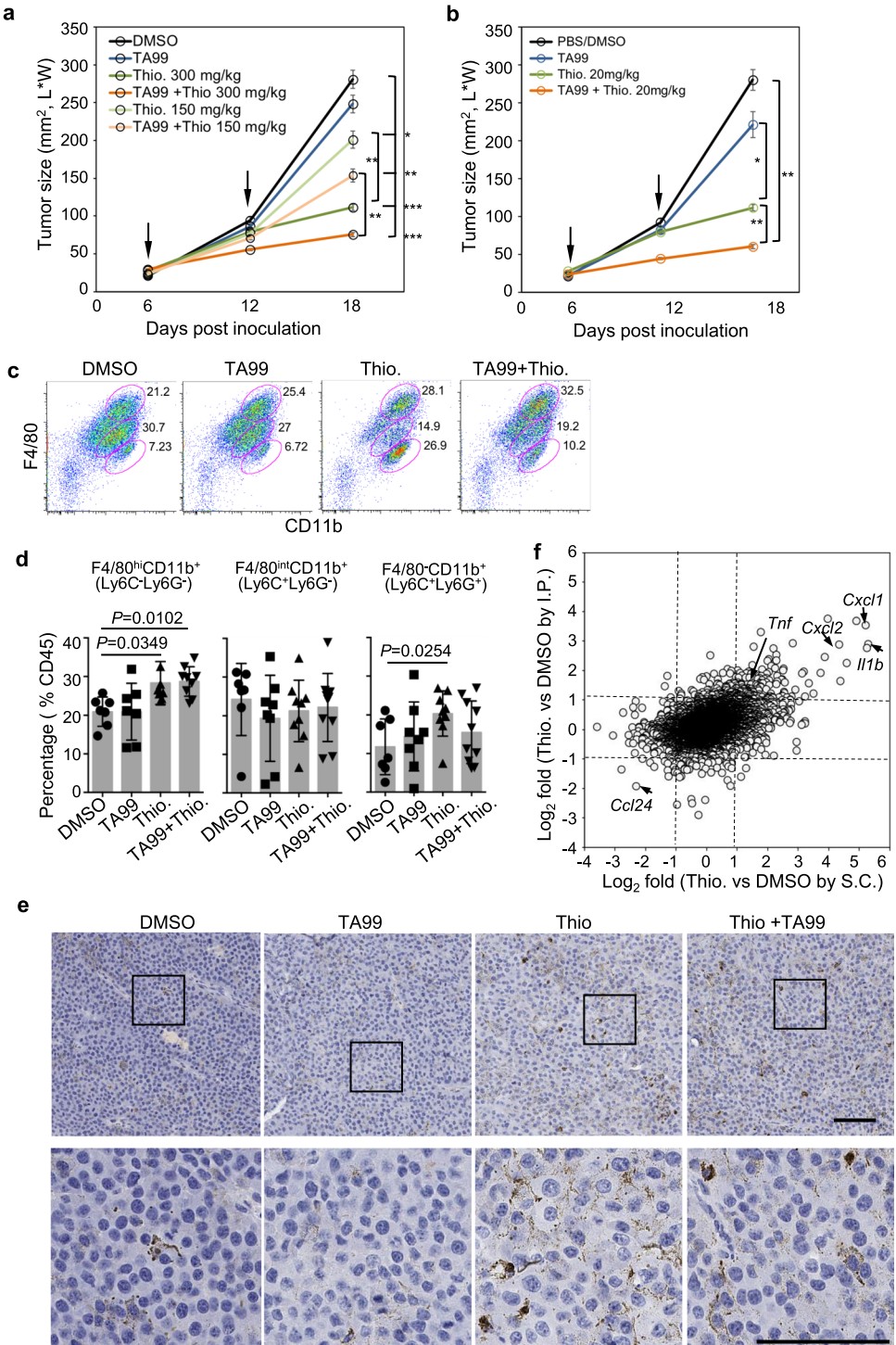

we identified known pathways, such as cytokine, in macrophage activation. More importantly, we identified many new pathways, including leptin, VEGF, EGF, and neurotransmitter, that mediate macrophage activation. Although studies have shown these pathways in macrophage function[39–43], their effects on macrophage activation and underlying mechanisms are unknown. Our transcriptional analysis of macrophages suggests that the ligands of these pathways activate macrophages by regulating gene expression of both typical M1 and M2 modules. For example, in hMDMs, leptin upregulates the expression of typical M1 modules induced by IFNγ while suppresses the expression of chronic inflammation TPP modules (Fig. 2). Notably, the ligands of

serotonin transporter and receptors, histamine transporter and receptors, dopamine transporter and receptors, and adrenoceptors all stimulate M1-like macrophage activation, suggesting the cross-talk between neuronal and immune systems and the potential roles of macrophage activation in neurological diseases[44].

Recent studies have shown that macrophages exhibit a multi-dimensional spectrum of phenotypes beyond M1 and M2[3,5,6,9]. Our identification of a diverse panel of macrophage-activating compounds that target GPCRs, enzymes, kinases, NHRs, and transporters (Fig. 1g) adds to the molecular basis of macrophage plasticity and further identifies new pathways in macrophage

**Fig. 6 Thiostrepton exhibits antitumor activities through reprogramming tumor-associated macrophages in vivo. a** Tumor growth curves in B6 mice bearing subcutaneous B16F10 tumors treated I.P. with DMSO, TA99, thiostrepton (150 or 300 mg/kg), and thiostrepton plus TA99 ($n = 6$–13 mice per group). **b** Tumor growth curves in B6 mice bearing subcutaneous B16F10 tumors treated I.P. with TA99, and S.C. with PBS or DMSO or thiostrepton (20 mg/kg) or thiostrepton plus TA99 ($n = 9$–11 mice per group). In **a** and **b**, arrows indicate dosing time points and data are presented as mean ± sem. *$P <$ 0.05, **$P < 0.01$ and ***$P < 0.001$ by two-sided $T$-test are indicated. **c, d** Flow cytometry analysis of TAM (F4/80$^+$CD11b$^+$Ly6C$^-$Ly6G$^-$), inflammatory monocytes (F4/80$^{int}$CD11b$^+$Ly6C$^+$Ly6G$^-$) and monocytes (F4/80$^-$CD11b$^+$Ly6C$^+$Ly6G$^+$) in the tumors from control, TA99-treated, thiostrepton-treated and thiostrepton plus TA99-treated tumor-bearing mice 18 days after tumor engraftment. Shown are representative F4/80 versus CD11b staining profiles gating on CD45$^+$ cells (**c**) and summarized data (mean ± sd) (**d**) from three independent experiments ($n = 7$–10 per group). **e** Representative immunohistochemistry staining with anti-F4/80 in tumor sections. The bottom panels are the enlargement of marked areas from the top panels. Brown: anti-F4/80 stain; blue: nuclear stain. Scale bar: 100 μm. Shown are representative staining from one mouse per group in **a**. **f** Comparison of gene expression changes induced by thiostrepton in tumor-infiltrating macrophages from individual mice following I.P. ($n = 4$) or S.C. administration ($n = 2$) of thiostrepton or DMSO ($n = 2$). Tumor infiltrated macrophages were sorted separately from tumor tissues of each mouse based on CD45$^+$F4/80$^+$CD11b$^+$Gr-1$^-$ 18 days after tumor engraftment, followed by separate RNA-seq. I.P. intraperitoneal injection, S.C. paratumor subcutaneous injection.

activation. Our extensive transcriptional analysis with over 40 selected compounds identifies how each compound stimulates macrophage activation through shared mechanisms and unique pathways. As expected, all compounds modulate macrophage activation through common pathways such as inflammatory response, immune response, and chemokine- and cytokine-mediated signaling pathways. Furthermore, each compound induces unique transcriptional responses of macrophages through their specific cellular targets, and many of these unique pathways are so far not known to mediate macrophage activation. For example, thiostrepton has been shown to have antiproliferative activity in cancer cells by inhibiting proteasome function or FOXM[29–31]. In both human and mouse macrophages, thiostrepton upregulates expression of proinflammatory genes, as well as genes associated with IFN/NFκB pathway and oxidative-reduction process (Fig. 5 and Supplementary Fig. 11). The transcriptional analysis also reveals that most of the identified compounds stimulate macrophage activation through modulating a fraction of M1- or M2-specific gene modules as well as common denominators that are induced by M1- or M2-activating cytokines (Fig. 4c, d). The milder effect is expected as individual compound only regulates specific signaling pathways through relevant protein targets (Fig. 4e). These observations further shed light on the nature of macrophage activation. The identified large panel of small molecule compounds and their corresponding targets and pathways are a rich resource for further studying basic macrophage biology.

Our study also provides a rich resource for exploring compounds/targets/pathways for modulating macrophage activation in disease intervention. Reprogramming macrophage has emerged as a significant approach for treating a variety of diseases[7–9]. Suppression or reprogramming of M2-like TAMs into M1-like macrophages by small molecule compounds is associated with induction of a strong antitumor activity alone or in combination with other therapeutics[14–17]. Similarly, suppression or reprogramming of M1-like macrophages into M2-like state significantly inhibits the progression of inflammatory and auto-immune diseases[45,46]. In this study, we confirm M1-activating compounds thiostrepton and cucurbitacin I potently reprogram TAMs toward M1-like macrophages and enhance antitumor activity either alone or in combination with an antibody therapeutic (Fig. 6 and Supplementary Figs. 12 and 13), suggesting that M1-activating compounds can be explored for reprogramming M2-like macrophages for the treatment of cancer and fibrosis where M2-like macrophages play a significant role in the disease processes. Similarly, M2-polarizing compounds can be explored for the treatment of inflammatory diseases by suppressing the inflammatory activities of M1-like macrophages. In complex diseases, pathogenic macrophages are known to be heterogeneous including both M1- and M2-like phenotypes or

have a transit, or intermediate phenotype with mixed characteristics of M1-like and M2-like phenotypes, or exhibit a dynamic phenotype during the disease progression[47–49]. To target the desired macrophage population, it is critical to suppress the expression of signature genes/pathways in the pathogenic macrophages at the correct time window. Our identification of unique pathways modulated by each compound provides a basis for selecting the appropriate compounds to reprogram macrophages for precision disease intervention.

## Methods

**hMDMs and cell lines.** Human peripheral blood mononuclear cells (PBMCs) were isolated from fresh blood, purchased from Research Blood Components, LLC, by density gradient centrifugation with Ficoll-Paque Plus (GE Healthcare) and Leucosep$^{TM}$ (Greiner Bio-One). Human monocytes were purified from PBMC using the EasySep$^{TM}$ Human Monocyte Isolation Kit (Stemcell Technologies) according to the manufacture's protocol. For in vitro differentiation of monocytes into human macrophages (M0, primary macrophage), isolated monocytes were cultured in complete RMPI1640 supplemented with 10% fetal calf serum (FCS) (Gibco), 2 mM L-glutamine (Corning) and 1% PenStrep solution (Corning) in the presence of 50 ng/mL recombinant human M-CSF (Peprotech) for 7 days. Tumor cell line B16F10 were purchased from ATCC and cultured in complete DMEM supplemented with 10% FCS, 1% PenStrep, and 2 mM L-glutamine. Human monocyte cell line (THP-1) was purchased from ATCC and cultured in complete RPMI 1640 containing 10% FCS, 2 mM L-glutamine, 1% nonessential amino acids (Lonza), 1 mM sodium pyruvate (Cellgro), and 1% PenStrep. Luciferase-expressing human lymphoma B cell line (GMB) was described previously[17] and cultured in complete RPMI 1640 containing 10% FCS, 2 mM L-glutamine, 0.55 mM 2-mercaptoethanol (Gibco), 1% nonessential amino acids (Lonza), 1 mM sodium pyruvate (Cellgro), and 1% PenStrep. All uses of human material have been approved by the Institutional Review Board at Massachusetts Institute of Technology (MIT).

**High-throughput compound screening, high-content microscope, and image analysis.** Based on the shape difference of M1 (round) and M2 (elongated) differentiated macrophages, we developed a high-throughput method to screen compounds which could modulate macrophage polarization. Human M0 primary macrophages differentiated from monocytes in vitro were seeded using a Multidrop Combi Dispenser (Thermo Scientific) at a density of 5000 cells/well in 50 μL complete RPMI in the presence of 10 ng/mL M-CSF into optical 384-well plates (Cat. 393562, BD Falcon) and cultured for 16 h for cell recovery. Around 20% of macrophages at this stage (M0) were elongated. Cells were treated with a library of 4126 compounds or drugs at the final concentration of 20 μM using the CyBi-Well simultaneous pipettor (CyBio). The compound library composes of 2066 bioactive compounds, 320 FDA-approved drugs, 440 oncological drugs, and 1280 natural compounds from the Center for the Development of Therapeutics at Broad Institute of MIT and Harvard. After 24 h incubation, supernatants were removed using the microplate washer (Bioteck) and cells were fixed by adding 50 μL 16% paraformaldehyde (Thermo Scientific) with the dispenser for 20 min. Cells were then washed with 50 μL 1xPBS twice and incubated for 20 min with NucBlue and AF746 Phalloidin (Invitrogen) to stain nucleus and cytoskeleton. Cells were then washed with 50 μL 1xPBS twice and maintained in PBS for image acquisition. Plates were read in the Opera Phenix high content screening system (PerkinElmer) to photograph cells using ×20 objective in two fluorescent channels (blue and far red). A total of six different fields in each well and an average of 1000 cells were imaged per well. CellProfiler[22] was used to identify each cell by overlapping signals from its nucleus and cytoskeleton, and calculate the eccentricity as the parameter to measure cell morphology. The Z-score was calculated by T-test to measure the difference of cell morphology between each treatment and control. The first and last two columns of each row of the 384-well plate were treated with the

same concentration of DMSO and combined as the control for the other 20 treatment wells in that row. For positive controls, classical M1 and M2 stimuli were added to each plate to generate the gold-standard Z-score cutoffs with M1 or M2 activation. M1 stimuli include LPS (100 ng/mL), IFNγ (50 ng/mL, Peprotech), TNFα (50 ng/mL, Peprotech), or IFNγ plus TNFα. M2 stimuli include IL-10 (10 ng/mL, Peprotech), IL-4 (10 ng/mL, Peprotech), or IL-13 (5 ng/mL, Peprotech). The gold-standard Z-scores were used as the cutoffs to identify compounds that activate macrophage into M1 or M2 state.

To screen for compounds which could reactivate or reprogram differentiated macrophages, 127 M1-activating and 180 M2-activating compounds from the first-round screen were cherry-picked. Human macrophages were seeded into optical 384-well plates for 16 h and then differentiated into either M1 macrophages in complete RPMI with 50 ng/mL IFNγ and 50 ng/mL TNFα or M2 macrophages in complete RPMI with 5 ng/mL IL-4 and 5 ng/mL IL-13. Twenty-four hours later, M1 macrophages were treated with M2-activating compounds and M2 macrophages were treated with M1-activating compounds for 24 h. Two independent experiments were performed with or without replacing differentiating medium right before treatment. Cell imaging and analysis were performed as above.

**Compound target and pathway analysis**. The identified compounds were classified based on the database from the International Union of Basic and Clinical Pharmacology (IUPHAR) (guidetopharmacology.org). The protein targets of the compounds were text-mined based on the target databases of UPHAR and DrugBank (drugbank.ca). The pathway enrichment analysis of protein targets of compounds was based on the WikiPathways[50].

**Mice, antibodies, flow cytometry, and western blotting**. Both C57BL/6 (B6) mice (stock #000664) and NOD.Cg-$Prkdc^{scid}$ $Il2rg^{tm1wjl}$/SzJ (NSG) mice (stock #005557) were purchased from the Jackson Laboratory and maintained in a specific pathogen-free animal facility at MIT under 12-h light dark cycles, controlled temperature (~23 °C), and 40–50% humidity with free access to food and water. All animal studies and procedures were approved by the MIT Committee on Animal Care (CAC). Experimental and control mice were from the same breeding colony and cohoused. Flow cytometry antibodies specific for mouse CD11b (M1/70), F4/80 (BM8,1:100), MHC⁻II (M5/114.15.2,1:100), Ly6C (HK1.4,1:100), Ly6G (1A8,1:100), Gr-1 (RB6-8C5,1:100), CD80 (16-10A1,1:100), CD86 (GL-1,1:100), CD163 (S15049I,1:100), CD206 (C068C2,1:100), IFNγ (XMG1.2,1:100), and TNFα (MP6-XT22,1:100) were from BioLegend (USA) and iNOS (CXNFT,1:100) from eBioscience (USA). Flow cytometry antibodies specific for human CD80 (2D10,1:50), CD86 (BU63,1:50), CD163 (GHI/61,1:50), and CD206 (15-2,1:50) were from BioLegend (USA) and iNOS (4E5,1:50) from Novus Biologicals (USA). Antibody ARG1 (A1exF5,1:100) specific for both human and mouse was from eBioscience (USA). B16F10 melanoma specific antibody TA99 for in vivo study was prepared as described[33]. Single-cell preparation from different organs, staining of cells with fluorophore-conjugated antibodies, and analysis of the stained cells using flow cytometry are as described[51]. Briefly, cells in single-cell suspension were incubated with specific antibodies at 4 °C for 20 min, washed twice, and resuspended in FACS buffer containing DAPI. Cells were fixed and permeabilized with Cyto-Fast Fix/Perm Buffer Set (BioLegend) for intracellular staining according to the manufacture's protocol. T cells were stimulated with the cell stimulation cocktail (eBioscience) for 4 h and then fixed/permeabilized for intracellular staining. Cells were run on BD-LSRII, collecting 20,000–100,000 live cells per sample. The data were analyzed by FlowJo.

Proteins were extracted from equal number of THP-1 cells with the CellLytic™ Lysis Reagent (Sigma). Samples containing 20 µg total protein (BCA™ Protein Assay Kit, Pierce Biotechnology) were resolved on a 10% SDS-PAGE gel and electro-transferred onto a PVDF membrane (Millipore Corporation). The membrane was blocked in 5% (w/v) fat-free milk in PBST (PBS containing 0.1% Tween-20). The blot was hybridized overnight with primary antibodies: anti-pSRC (D49G4, Cell signaling technology, 1:1000) and pSIK1/2/3 (#ab199474, Abcam, 1:1000) according to the recommended dilution in 5% fat-free milk. The blot was washed twice in PBST and then incubated with anti-Rabbit HRP-conjugated secondary antibody (Cell Signaling Technology, 1:2000) in 5% fat-free milk. The membrane was washed twice in PBST and subjected to protein detection by ECL Plus Western Blotting Detection System (GE Healthcare) before being exposed to a Kodak BioMax XAR film. The membrane was stripped and reblotted with the anti-β-tubulin (D49G4, Cell Signaling Technology) for protein loading control.

**Mouse tumor model and treatment**. For the melanoma model, an inoculum of $1 \times 10^6$ B16F10 tumor cells was injected subcutaneously on the flank of 8–10-week-old male B6 mice in 100 µL sterile PBS. Six days following tumor inoculation, mice were randomized into four treatment groups: control (PBS or DMSO), tumor-targeting antibody TA99, compound, and compound plus TA99. TA99 was administered at 100 µg per dose intraperitoneally. The compound was administrated at the indicated dosage by either I.P. or paratumor S.C. injection. All mice were dosed at day 6 and day 12 post tumor inoculation for a total of two treatments and monitored daily. Tumor size was measured as an area (longest dimension × perpendicular dimension) at day 6, day 12, and day 18 post tumor inoculation.

Mice were euthanized by $CO_2$ narcosis for analyses at day 18 post tumor inoculation before tumors reach the maximal size of 500 mm² allowed in our approved MIT CAC protocol. For the lymphoma model, $1 \times 10^7$ GMB cells was injected through tail vein in 100 µL sterile PBS into 10–12-week-old male NSG mice. Mice were treated 2 weeks post tumor cell engraftment. Tumor-targeting antibody rituximab (InvivoGen) was administered at 10 mg/kg intraperitoneally. The compound was administrated I.P. at the indicated dosage. All mice were dosed at week 2 and week 3 post tumor injection for a total of two treatments and monitored daily. Tumor growth and spread was visualized using an IVIS Spectrum-bioluminescent imaging system (PerkinElmer) at weeks 2, 3, and 4 post tumor injection. Images were analyzed with the Living Image Software (PerkinElmer). Mice were euthanized by $CO_2$ narcosis for analysis at week 4 post tumor inoculation

**Histopathology and immunochemical staining**. Mice were euthanized and tumor tissues were harvested and fixed with 10% neutral-buffered formalin solution (Sigma-Aldrich) for 24 h. The tissues were processed with Tissue Processor (Leica Microsystems) and embedded in paraffin. Sections were cut at 5 µM thickness, mounted on polylysine-coated slides (Thermo Fisher Scientific), dewaxed, rehydrated, and processed for hematoxylin and eosin staining according to a standard protocol. For immunochemical staining, antigen retrieval was carried out by microwaving the slides in 0.01 M sodium citric acid buffer (pH 6.0) for 30 min. Sections were then immersed for 1 h in blocking buffer (3% BSA, 0.2% Triton X-100 in PBS), then incubated in primary antibody in blocking buffer at 4 °C overnight, followed by incubation with secondary antibody conjugated HRP at 4 °C for 1 h. Hematoxylin was used as the nuclear counterstain. Stained tumor sections were scanned with a high-resolution Leica Aperio Slide Scanner. Images were analyzed by Aperio ImageScope software.

**Mouse bone marrow-derived macrophages (mBMMs) and TAMs**. mBMMs were prepared as described previously[52]. Briefly, fresh bone marrow cells were isolated from B6 mice, plated into a six-well plate with $1 \times 10^6$/mL in complete RPMI with 2-mercaptoethanol and cultured for 6 days with medium change every 2 days. mBMMs were differentiated to resemble TAMs in the presence of 10 ng/mL mIL-4 and mIL-13 (Peprotech) or 25 mM lactic acid[53] or tumor CM for 24 h. To prepare CM, 70% confluent B16F10 cultures were replaced with fresh medium and the tumor medium was collected 24 h later and filtered (0.2 µm). The mixture of three volumes of tumor medium with one volume of complete RMPI for mBMM serves as the CM. Expression of Arg, Fizz1, and Vegfa were quantified by qPCR to assess the development of TAMs. Expression of Tnf, Il1b, Nos2, Cxcl2, Ccl5, Ym1, and Tgfb served as macrophage activating markers. To assay the tumor growth inhibition, mBMMs (10,000 cells per well in a 96-well plate) were treated with thiostrepton for 24 h and then cocultured with equal number of B16F10 melanoma cells in fresh complete RPMI for 12 h. Tumor cells were quantified by flow cytometry in the coculture. In addition, the conditioned media from mBBM cultures treated or not treated with thiostrepton were collected and filtered with or without heat inactivation at 95 °C for 5 min. B16F10 melanoma cells were cultured with the CM for 12 h and the surviving tumor cells were quantified by flow cytometry.

**RNA isolation, sequencing, and data analysis**. RNAs were extracted with RNeasy MinElute Kit (Qiagen), converted into cDNA and sequenced using Next-Generation Sequencing (Illumina). RNA-seq data were aligned to the mouse genome (version mm10) and raw counts of each genes of each sample were calculated with bowtie2 2.2.3[54] and RSEM 1.2.15[55]. Differential expression analysis was performed using the program edgeR at $P < 0.05$ with a two fold-change[56]. The gene expression level across different samples was normalized and quantified using the function of cpm. DEGs were annotated using online functional enrichment analysis tool DAVID (http://david.ncifcrf.gov/)[57]. Gene set enrichment analysis were performed with GSEA[58] with FDR q-value < 0.05. The heatmap figure was visualized with MeV[59]. To quantify the levels of RNA transcripts, total RNA was extracted from various cells and reverse transcribed by TaqMan™ Reverse Transcription Reagents Kit (ABI Catalog No. N8080234), followed by amplification with SYBR Green Master Mix (Roche Catalog No. 04707516001) with specific primers (Supplementary Table 3) and detected by Roche LightCycler 480. The Ct values were normalized with housekeeping gene GAPDH for comparison.

**Macrophage activation network analyses**. To determine the central hubs of the core macrophage activation network induced by compounds, transcriptional interactions between genes were first determined by ARACNe[23] based on the perturbed transcriptional profiles of 34 compounds as well as IFNγ and IL-4 controls. The 12549 expressed genes were taken into calculation of mutual information with $P$ value less than 1e−7. The threshold of the data processing inequality theorem from information theory used by ARACNe was set to 0.1 to detect total 400,165 regulatory interactions in the core macrophage activation network. GO enrichment analysis and enrichment map of top 10% central hubs (1255 genes) was performed by BiNGO[27]. The network was visualized by Cytoscape[60].

**Statistic methods**. Statistical significance was determined with the two-sided unpaired or paired Student's $t$ test. The FDRs were computed with $q = P \times n/i$, where $P = P$ value, $n =$ total number of tests, and $i =$ sorted rank of $P$ value.

**Reporting summary**. Further information on research design is available in the Nature Research Reporting Summary linked to this article.

## Data availability

Raw RNA-seq data are deposited in the database of Gene Expression Omnibus with accession IDs: GSE144992 and GSE155551. Compound-screening data and DEGs are provided as Supplementary Data files. All other data that support the findings of this study are available from the corresponding authors upon reasonable request. Certain databases of IUPHAR and DrugBank used for text mining the information of compounds are accessible via guidetopharmacology.org and drugbank.ca. Source data are provided with this paper as a Source Data file.

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

## Acknowledgements

The authors would like to thank Starsha Kolodziej and Megan Kaiser for technique assistance, the Center for the Development of Therapeutics at the Broad Institute of MIT and Harvard for the initial screen, and Koch Institute Swanson Biotechnology Center and core facilities for assistance with the secondary screen, flow cytometry, histology, and RNA-seq data acquisition and analysis. The authors would also like to thank members of Chen Lab for their suggestions. This work was supported in part by National Institutes of Health Grants CA197605 and NS104315, Ivan R. Cottrell Professorship and Research Fund, the Koch Institute Support (core) Grant P30-CA14051 from the National Cancer Institute, and the National Research Foundation of Singapore through the Singapore–MIT Alliance for Research and Technology's Interdisciplinary Research Group in Antimicrobial Resistance Research Program.

## Author contributions

G.H. and J. Chen designed the research, interpreted the data, and wrote the manuscript. G.H., Y.S. and D.R.B. performed the initial screening. G.H. and J. Cheah performed the further screening. G.H. and B.H.K. performed experiments related to melanoma studies. Z.F. and T.D. performed the fluorescence microscopy analysis. G.H. performed all other experiments. G.H., J. Chen, and K.D.W. analyzed the data.

## Competing interests

J. Chen and G.H. (inventors) declare that a provisional patent application related to this work has been filed with the United States Patent and Trademark Office on September 21, 2020. The other authors declare no competing interest.
