## [Peer Review File · Nature Communications]

REVIEWER COMMENTS

Reviewer #1 (Immune metabolism, M1/M2 macrophage)(Remarks to the Author):

In this manuscript, Hu et al. developed a high-throughput screening method to examine the immunomodulatory potential of a collection of 4126 chemicals containing FDA-approved drugs/bioactive molecules. The system was based upon a well-characterized observation that M1-polarized and M2-polarized macrophages display different cytologic morphologies. Using this imaging-based screening, the authors identified several compounds that polarized macrophages towards the M1 or M2 phenotype. RNA-seq analyses were subsequently performed to determine the gene pathways affected by compounds from their screen. Finally, the authors examined one M1-polarizing compound, thiostreptin, which they showed to be effective in limiting tumor size in a mouse melanoma model. This study is not particularly novel, nor does it shed new light on macrophage or tumor biology; however, it does serve as a strong proof-of-concept approach for potential repurposing of current approved drugs for immunotherapy.

Major Comments:

1. RNA-seq was used extensively to characterize the drug effects on macrophage activation. However, information related to the sample size/biological replicates was missing (this applies to all other assays as well). More detailed statistical analyses of the RNA-seq results should also be provided, particularly if $n=1$. Similarly, the age and sex of mice for the tumor study should be included.
2. Tumor-associated macrophages may be characterized as M2-like, but they are not M2. Fig. 5d should be repeated using tumor-conditioned medium to promote TAM polarization. In addition, the tumor killing activity is primarily from T cells, not macrophages. As such, Fig. 5e was not a valid approach. Since the main claim is thiostreptin promotes anti-tumor immunity, the evidence supporting this claim needs to be strengthened.
3. The conclusions of this paper rest almost entirely on data from RNA seq. The authors need to validate some of the major genes with qPCR.
4. In Fig. 6a/b, the possibility that thiostreptin has a direct effect on tumor progression can not be ruled out. Minimally, authors need to examine Arg1 and other markers in TAM isolated from primary tumors by flow cytometry. It would also be informative to determine IFN γ producing T/NK cells to assess the anti-tumor immunity in vivo.

Minor comment

1. In the Introduction, the authors stated that M2 macrophages produce IL-4 and IL-13, which is incorrect.

Reviewer #2 (Cancer immune therapy, tumor microenvironment)(Remarks to the Author):

In the manuscript entitled "High throughput phenotypic screen and transcriptional analysis identify new compounds, targets and pathways for macrophage reprogramming" Hu and colleagues use a phenotypic/morphological screen in order to identify novel compounds able to reprogram macrophages to classically or alternatively activated states. Macrophage activation, whether classical or alternative, is implicated in many diseases including cancer. In several cancer models, reprogramming TAMs to classically activated types can be a good therapeutic strategy to aid "standard of care" or immune checkpoint blockade. Therefore, developing screening strategies to identify compounds that can reprogram macrophage activation states is of high translational relevance. The present study initially identified ~300 compounds able to shift the polarization state of macrophages in vitro. This was then validated through RNA-seq of non-redundant hits to identify common pathways modified by the compounds. Finally, one of the "M1-activating" compounds, thiostreptin was used to validate the activity in reprogramming activation state of macrophages in a mouse tumor model for its

use as an anti-cancer therapy.

While the validation used for the initial screening was a simplified way of looking at macrophage activation/polarization state, the fact that the authors were able to identify novel compounds able to induce anti-tumor microenvironments is of high translational relevance to develop novel therapies for various inflammatory diseases as well as cancer. However, this novel assay design to screen compounds for reprogramming macrophage activation states lacks important quality control to ascertain the cellular morphologies associated with M1/M2. Also, tumor model studies need to be more detailed and explained with much more detail in the results and discussion.

Major revision:

1. A major issue with the manuscript is its reliance on morphology of macrophages alone. While macrophages do show distinct morphology tightly connected to known activating signals, the authors must also include bonafide markers to ascertain M1/M2 states prior to moving into the morphology-based screening. In particular, gene expression (Taqman of genes associated with activation state from the literature), as well as flow cytometry of the baseline conditions (for instance IFN γ and IL-4 activated), would strengthen the morphology-based detection of macrophage activation specially when going into actual screen of compounds unknown for their impact on macrophage activation. M1 and M2 specifically are defined as metabolically-skewed states utilizing arginine by iNOS (M1) or arginase (M2); the authors must at least analyze expression of iNOS in addition to arginase expression. This must also be clear in the text, as the screening is more comparable to IFN γ and IL-4 activated macrophages *in vitro*, and not necessarily to what one would find *in vivo*. This issue is partly reflected by the fact that only reduction of M1-like markers, with no changes in IL-4 signaling, is observed when treated with their M2 compounds (p7, line 177). This again sheds light on the fact that one has to be careful when designating macrophage polarization state/programming since by definition reprogramming or polarization is not just loss of certain traits but also acquisition of other features.
2. The manuscript currently uses RNAseq and the observation of altered cytoskeletal as proof of concept for morphology-based screening (line 370-373). While it is encouraging to observe these changes in the RNAseq data after exposure to their novel agents, other methods to validate macrophages activation must accompany Fig 1 and Fig. 2.
3. In Fig. 3, macrophages are pre-stimulated with selected cytokines prior to being treated with compounds. Importantly, the cytokines are withdrawn prior to treatment. Macrophages in these cultures would be driven/dependent on these extracellular stimuli, and withdrawal of the signal prior to adding the "reprogramming" compound, which in turn could affect the experimental output.
4. Authors should validate the effect of 17 M1/M2 reprogramming compounds for their effect on changes on Arg and iNOS expression to provide strength to author's claim that these compounds can alter M1 state to M2 or vice-versa.
5. Authors show in a co-culture of macrophages and B16F10 cells that thiostrepton activates and reprograms macrophages toward a pro-inflammatory state and enhances their tumoricidal activity *in vitro*. Author should perform this experiment in a quantitative setting and also look for definitive tumor cell killing, if at all, by thiostrepton treated macrophages in co-culture. This would be helpful in explaining results in fig.6.
6. While it is interesting to see decreased tumor growth, the exact state of the tumor-associated macrophages after treatment is not clear. These macrophages should be assessed with some activation markers in the flow cytometry data shown in Fig. 6d (like MHCII, CD206, CD86 mentioned in material and methods). This would allow identification whether thiostrepton is able to alter the activation state of TAMs in this mouse model.
7. Given the role of M1 macrophages in increasing adaptive immune response, which seems to be crucial in cancers with high mutation loads such as melanoma the adaptive immune response e.g., CD8 T cell frequencies and their activation in thiostrepton treated cohorts should be analyzed to strengthen these impressive *in vivo* results.
8. Importantly, since thiostrepton could be acting on numerous cell types *in vivo*, a macrophage

depletion strategy must be utilized to discern the contribution of macrophages in vivo to these results. 9. Also implies by the findings is relief of macrophage-mediated T cell suppression. Cd4 and CD8 depletion studies would need to be conducted to provide this data in support of the authors conjecture.

Minor revision:

1. The selection criteria for picking thiostrepton over for instance mocetinostat is not clear to the reviewer, as the latter seem to induce IFN γ signaling resembling recombinant IFN γ in vitro Fig. 2d. Authors should explain rationale for choosing thiostrepton, if any.
2. The authors use the term TAMs for mouse bone marrow macrophages exposed to IL4 and IL13, which would be incorrect as these are quintessential M2 as per also described at multiple points by authors in the manuscript. This need to be corrected to M2 macrophages, and referred to as bone marrow-derived macrophages (BMdM)

Reviewer #3 (M1/M2 macrophage, inflammation)(Remarks to the Author):

The manuscript written by Dr. Hu et al. shows that they developed a technique to screen various chemicals and drugs' potentials to induce M1 or M2 macrophage phenotypes by analyzing changes of cell morphology. I feel this work is potentially interesting, however, there are some concerns.

<Major remarks>

1)

Authors certainly show that picked up compounds could change gene expressions of macrophages, however, in the current version there are few data to address whether the molecules are changed in protein level, and where the point of action of the drug is. Regarding Bosutinib and Thiostrepton, authors should clarify the key molecules affected by each drug in the macrophage functional changes. By doing so, the reliability of the data will increase, which in turn would increase the reliability of this screening method.

1-a) Is the tyrosine kinase that is the target of Bosutinib known generally as key molecules in macrophage polarization?

1-b) If so, please show that Bosutinib effect on the activation of the kinase (ex. phosphorylation), and the inhibitor of the kinase can cancel the effect of Bosutinib and M2 polarization.

1-c) As the same concept as 1-a) and 1-b), what molecule is the target of Thiostrepton? Is the molecule essential for M1 polarization?

2)

In the in vivo tumor model in Fig. 6, Thio shows the antitumor effect, and it can also activate M1 macrophages. With these two data, the authors state that Thio induces M1 macrophages in vivo, which is effective against the tumor. However, as the authors discussed, there is a possibility that Thio suppresses tumor growth regardless of macrophage activation. To verify this, I recommend additional experiments.

2-a) Thio-primed macrophages are injected into the tumor and see the anti-tumor effects.

2-b) Macrophages are depleted by liposomes or other reagents, then does the anti-tumor effect of Thio disappear?

<Minor remarks>

In Fig 1a and 1b, Z-score of M1 cells is negative ($Z=-xx$). On the other hand, in Fig 1e, Z-score of Compound 1, which can induce M0 to M1, is positive ($Z=12$). I have read the figure legend, Fig 1e shows the difference in cell morphologies comparing to DMSO control, but I could not understand why the Z score of compound 1 move positive compared with DMSO. I guess I've misunderstood something, please expand results or legends so that the reader can easily understand them.

REVIEWER COMMENTS

Reviewer #1 (Immune metabolism, M1/M2 macrophage)(Remarks to the Author): In this manuscript, Hu et al. developed a high-throughput screening method to examine the immunomodulatory potential of a collection of 4126 chemicals containing FDA-approved drugs/bioactive molecules. The system was based upon a well-characterized observation that M1-polarized and M2-polarized macrophages display different cytologic morphologies. Using this imaging-based screening, the authors identified several compounds that polarized macrophages towards the M1 or M2 phenotype. RNA-seq analyses were subsequently performed to determine the gene pathways affected by compounds from their screen. Finally, the authors examined one M1-polarizing compound, thiostreptin, which they showed to be effective in limiting tumor size in a mouse melanoma model. This study is not particularly novel, nor does it shed new light on macrophage or tumor biology; however, it does serve as a strong proof-of-concept approach for potential repurposing of current approved drugs for immunotherapy.

Response: We thank the reviewer for the constructive comments and have revised the manuscript accordingly to address the following comments.

Major Comments:

1. RNA-seq was used extensively to characterize the drug effects on macrophage activation. However, information related to the sample size/biological replicates was missing (this applies to all other assays as well). More detailed statistical analyses of the RNA-seq results should also be provided, particularly if n=1. Similarly, the age and sex of mice for the tumor study should be included.

Response: For RNA-seq studies with 8 selected compounds in Figure 2 and 34 selected compounds in Figure 4, two replicates were used per compound. For RNA-seq studies with macrophages purified from tumors in Figure 5, macrophages were sorted from the tumor of individual mice with 2-4 mice per group and then subjected to RNA-seq separately. We have included the sample sizes for RNAseq and other assays in the figure legends of the revised manuscript. Because B16F10 tumor line was derived from a male C57BL/6 mouse, male C57BL/6 mice at 8-10 weeks of age were used. Male NSG mice at 10-12 weeks of age were used. The information on sex and age of the mice are included in Materials section of the revised manuscript.

2. Tumor-associated macrophages may be characterized as M2-like, but they are not M2. Fig. 5d should be repeated using tumor-conditioned medium to promote TAM polarization. In addition, the tumor killing activity is primarily from T cells, not macrophages. As such, Fig. 5e was not a valid approach. Since the main claim is thiostreptin promotes anti-tumor immunity, the evidence supporting this claim needs to be strengthened.

Response: Per reviewer's suggestion, we examined the effects of thiostrepton on the polarization of TAM generated from B16F10-derived tumor-conditioned medium.

Consistent with original Fig. 5d, thiostrepton inhibited the TAM function by suppressing the expression of *Arg1*, *Fizz1*, *Vegfa* and M2 markers *Ym1* and *Tgfb*, but promoted the expression of M1 markers *Il1b*, *Tnf*, and *Cxcl2* as well as *iNos* (*Nos2*) as measured by qPCR (revised Fig. 5d). Moreover, at the protein level thiostrepton suppressed Arg1 in TAM but up-regulated M1-activating makers MHCII, CD80 and iNOS (Supplementary Fig. 7a). These results were included in the revised manuscript as Fig.5d and Supplementary Fig. 7.

In Fig. 5e, macrophages were either treated or not treated with thiostrepton for 24 hrs. After removing thiostrepton from the medium, the primed macrophages as well as untreated control macrophages were co-cultured with tumor cells to determine the effect of macrophages on tumor cell growth. Compared to untreated macrophages, macrophages activated by thiostrepton inhibited the tumor cell growth in a thiostrepton dose-dependent manner. Also we examined the effects of conditioned medium on tumor cell growth from macrophages either treated or not treated with thiostrepton for 24 hrs. As shown in the Supplementary Fig.8a, mouse BMMs were treated with thiostrepton for 24 hrs (same as Fig. 5e), conditioned medium was collected and filtered. The same numbers of B16F10 melanoma cells were cultured for 12 hrs with conditioned medium or conditioned medium heated at 95°C for 5 min. As expected, the medium from macrophages treated with thiostrepton inhibited the tumor cell growth compared to medium from untreated macrophages. But heat-inactivated conditioned medium did not inhibit the tumor cell growth. These results suggest that thiostrepton treatment of macrophages induces the production of a tumor growth inhibitor that is heat-labile.

It is true that tumor killing activity is primarily mediated by T cells, not macrophages, in mice and humans. However, macrophages could kill tumor cells through phagocytosis, especially antibody-dependent cellular phagocytosis (ADCP). To strengthen our conclusion, we performed ADCP with human blood monocyte-derived macrophages (hMDM) and mouse bone marrow-derived macrophage (mBMM) with and without thiostrepton treatment. We used CD20+ B-cell lymphoma cells and Rituximab (anti-CD20) to assess the ADCP activity. Briefly, tumor cells were labeled with eFluro670 and Ritumimab, then co-cultured for 2 hrs with hMDM or mBMM that have been primed with thiostrepton for 24 hrs. Macrophages with engulfed tumor cells (eFluro670+) among total macrophages (CD14+) were quantified by flow cytometry. Results showed that ~12% more thiostrepton-treated hMDM engulfed tumor cells than DMSO-treated hMDM (67.8% vs. 55.6%), while ~4% more thiostrepton-treated mBMM engulfed tumor cells than DMSO-treated mBMM (30.3% vs. 26.5%). These results were included in the revised manuscript as supplementary Fig. 8b,c.

3. The conclusions of this paper rest almost entirely on data from RNA seq. The authors need to validate some of the major genes with qPCR.

Response: Per reviewer's suggestion, we performed qPCR to validate some of major genes observed in RNAseq data. We extracted the expression changes of typical M1 and M2 markers whose expression was altered by the selected compounds based on RNAseq results as described in Fig. 4. We identified 13 M1 marker genes that were up-regulated

by the 17 selected M1-activating compounds but down-regulated by the 17 selected M2-activating compounds, as well as 15 M2 marker genes that were up-regulated by the M2-activating compounds but down-regulated by the M1-activating compounds (Supplementary Fig. 5c). We selected 4 M1 genes CD86, TNF, IL1B and CXCL2 and 3 M2 genes CD163, CD206 (MRC1) and IL10 for validation in hMDM by qPCR. Results showed that the transcript levels of M1 genes were up-regulated by M1-activating compounds while those of the M2 genes were up-regulated by M2-activating compounds (Supplementary Fig. 5d). Consistently, by quantifying the protein levels by flow cytometry, CD80 and CD86 (M1 markers) were up-regulated by M1-activating compounds and down-regulated by M2-activating compounds, while CD163 and CD206 (M2 markers) were up-regulated by M2-activating compounds and down-regulated by M1-activating compounds (Supplementary Fig. 5e). Moreover, the M1 marker iNOS was upregulated by some M1-activating compounds but suppressed by most of M2-activating compounds. The M2 marker ARG1 was suppressed by M1-activating compounds. These results validate some of the major genes from RNA-seq and were included as the Supplementary Fig. 5c-e of the revised manuscript.

4. In Fig. 6a/b, the possibility that thiostreptin has a direct effect on tumor progression can not be ruled out. Minimally, authors need to examine Arg1 and other markers in TAM isolated from primary tumors by flow cytometry. It would also be informative to determine IFN γ producing T/NK cells to assess the anti-tumor immunity in vivo.

Response: We thank the reviewer for the excellent suggestion. We have prepared single cell suspension of tumors harvested from various mice at 18 days after tumor engraftment and analyzed Arg1, iNos (Nos2), IFN γ , TNF α expression by TAM, NK cells and/or T cells by flow cytometry. Results showed that Arg1 expression was significantly reduced in TAM from mice following treatment with thiostrepton alone or thiostreptin plus TA99, while the levels of iNos and CD86 in macrophages were significantly increased in mice following treatment with thiostrepton alone or thiostreptin plus TA99 (Supplementary Fig. 10a-b). Similarly, thiostrepton alone or thiostreptin plus TA99 also significantly increased IFN γ and TNF α expression by tumor-infiltrating NK cells but not CD8+ T cells (Supplementary Fig. 10c). These results are included in the revised manuscript as Supplementary Fig. 10.

Minor comment

1. In the Introduction, the authors stated that M2 macrophages produce IL-4 and IL-13, which is incorrect.

Response: We have addressed the reviewer's concerns by changing "IL4, IL13" to TGF β .

Reviewer #2 (Cancer immune therapy, tumor microenvironment)(Remarks to the Author):

In the manuscript entitled "High throughput phenotypic screen and transcriptional analysis identify new compounds, targets and pathways for macrophage

reprogramming” Hu and colleagues use a phenotypic/morphological screen in order to identify novel compounds able to reprogram macrophages to classically or alternatively activated states. Macrophage activation, whether classical or alternative, is implicated in many diseases including cancer. In several cancer models, reprogramming TAMs to classically activated types can be a good therapeutic strategy to aid “standard of care” or immune checkpoint blockade. Therefore, developing screening strategies to identify compounds that can reprogram macrophage activation states is of high translational relevance. The present study initially identified ~300 compounds able to shift the polarization state of macrophages in vitro. This was then validated through RNA-seq of non-redundant hits to identify common pathways modified by the compounds. Finally, one of the “M1-activating” compounds, thiostrepton was used to validate the activity in reprogramming activation state of macrophages in a mouse tumor model for its use as an anti-cancer therapy. While the validation used for the initial screening was a simplified way of looking at macrophage activation/polarization state, the fact that the authors were able to identify novel compounds able to induce anti-tumor microenvironments is of high translational relevance to develop novel therapies for various inflammatory diseases as well as cancer. However, this novel assay design to screen compounds for reprogramming macrophage activation states lacks important quality control to ascertain the cellular morphologies associated with M1/M2. Also, tumor model studies need to be more detailed and explained with much more detail in the results and discussion.

Response: We thank the reviewer for the constructive comments and have revised the manuscript accordingly to address the following comments.

Major revision:

1. A major issue with the manuscript is its reliance on morphology of macrophages alone. While macrophages do show distinct morphology tightly connected to known activating signals, the authors must also include bonafide markers to ascertain M1/M2 states prior to moving into the morphology-based screening. In particular, gene expression (Taqman of genes associated with activation state from the literature), as well as flow cytometry of the baseline conditions (for instance IFN γ and IL-4 activated), would strengthen the morphology-based detection of macrophage activation specially when going into actual screen of compounds unknown for their impact on macrophage activation. M1 and M2 specifically are defined as metabolically-skewed states utilizing arginine by iNOS (M1) or arginase (M2); the authors must at least analyze expression of iNOS in addition to arginase expression. This must also be clear in the text, as the screening is more comparable to IFN γ and IL-4 activated macrophages in vitro, and not necessarily to what one would find in vivo. This issue is partly reflected by the fact that only reduction of M1-like markers, with no changes in IL-4 signaling, is observed when treated with their M2 compounds (p7, line 177). This again sheds light on the fact that one has to be careful when designating macrophage polarization state/programming since by definition reprogramming or polarization is not just loss of certain traits but also acquisition of other features.

Response: We share the reviewer's concerns as M1 and M2 polarization itself has been controversial. In the first paragraph of "Discussion", we discussed in detail the rationale that cell shape change is a valid phenotypic profiling of macrophage activation with four lines of evidences. Previous studies have shown that both mouse and human macrophages exhibit dramatically different cell shapes following activation into different phenotypes *in vitro*: an elongated shape for M2-like macrophages and round shape for M1-like macrophages (Ref. 19, 22), based on cytokine profiles (Ref. 19, 38), the genome-wide gene expression (Ref. 20, 21) as well as the expression of widely used markers such as CD80, CD86, CD163 and CD206. Consistently, our RNAseq results showed IFN γ polarized hMDM into M1 while IL4 polarized hMDM into M2 phenotype. Per reviewer's suggestion, we have now quantified the levels of typical M1 and M2 markers at protein level by flow cytometry. Expression of M1 markers including HLA-DRB, CD80 and CD86 were up-regulated by IFN γ and suppressed by IL4 while M2 markers CD206 and CD163 were up-regulated by IL4 and suppressed by IFN γ . These results are included in the revised manuscript as Supplementary Fig. 1b.

We also performed further validations by qPCR at RNA level and by flow cytometry at protein level in the hMDM following compound treatment. At transcriptional level, the 17 M1-activating compounds stimulated expression of M1 marker genes, including CD86, TNF, IL1B and CXCL2, whereas the 17 M2-activating compounds stimulated expression of M2 genes, including CD163, CD206 and IL10 (Supplementary Fig. 5d). Consistently, at the protein level, CD80 and CD86 were up-regulated by M1-activating compounds but down-regulated by M2-activating compounds, whereas CD163 and CD206 were up-regulated by M2-activating compounds but down-regulated by M1-activating compounds (Supplementary Fig. 5e). Moreover, M1 marker iNOS was upregulated by some M1-activating compounds (4 out of 17) but suppressed by most of M2-activating compounds (14 out of 17). M2 marker ARG1 was up-regulated by 4 out of 17 M2-activating compounds, but suppressed by most 15 out of 17 M1-activating compounds. These results were included in the revised manuscript as Supplementary Fig. 5c-e.

We agreed with the reviewer's point that polarization is not just loss of certain traits but also acquisition of other features. Indeed, M1-compounds induced both gain and loss of modules similar to those induced by IFN γ and LPS, while M2-compounds induced more profound loss of M1-modules as well as gain of the LPS-suppressed modules (Fig. 4c). As we discussed, unlike strong cytokine stimuli of IFN γ or IL4, the effect of the compounds is more moderate on the global transcription of macrophage activation based on numbers of altered genes and pathways. These compounds only induced macrophages to gain and lose some of the modules as compared to cytokines.

Per reviewer's suggestion, in the revised manuscript we clearly stated that our compound modulation of macrophage activation was more similar to IFN γ and IL-4 activated macrophages *in vitro* in the results and discussion sections.

2. The manuscript currently uses RNAseq and the observation of altered cytoskeletal as proof of concept for morphology-based screening (line 370-373). While it is encouraging to observe these changes in the RNAseq data after exposure to their novel agents, other methods to validate macrophages activation must accompany Fig 1 and Fig. 2.

Response: Per reviewer's suggestion, we validated the protein expression changes of typical activation markers of M1 (CD80, CD86) and M2 (CD206, CD163) by flow cytometry in hMDM treated with compounds. All six M1-activating compounds upregulated expression of M1 markers CD80 and CD86 and down-regulated expression of M2 markers CD163 and CD206. Both M2-activating compounds down-regulated M1 markers and up-regulated M2 marker CD163. These results are included in the revised manuscript as Supplementary Fig. 3c.

In line with these results, we performed further validations at both RNA level by qPCR and protein level by flow cytometry after treating already polarized hMDM with compounds. At transcriptional level, M1 genes (CD86, TNF, IL1B and CXCL2) were up-regulated by M1-activating compounds while M2 genes (CD163, CD206 and IL10) were up-regulated by M2-activating compounds (Supplementary Fig. 5d). Consistently, at the protein level, CD80 and CD86 were up-regulated by M1-activating compounds and down-regulated by M2-activating compounds, while CD163 and CD206 were up-regulated by M2-activating compounds and down-regulated by M1-activating compounds (Supplementary Fig. 5e). These results are included in the revised manuscript.

3. In Fig. 3, macrophages are pre-stimulated with selected cytokines prior to being treated with compounds. Importantly, the cytokines are withdrawn prior to treatment. Macrophages in these cultures would be driven/dependent on these extracellular stimuli, and withdrawal of the signal prior to adding the "reprogramming" compound, which in turn could affect the experimental output.

Response: We performed experiments in both the absence and presence of polarizing cytokines. Fig. 3a and b are results from cytokine withdrawn prior to compound treatment and Fig. 3d and e are results from compound treatment in the continuous presence of cytokines. Surprisingly, more compounds exhibited significant effects on cell shape changes in the presence of the polarizing cytokines (67 M1- and 55 M2-activating) than in absence of the polarizing cytokines (46 M1- and 25 M2-activating) at the same compound concentration of 5 μ M. Most importantly, ~80% of compounds

overlapped in both settings. As we discussed, the presence of differentiating cytokines appears to make macrophages more sensitive to reprogramming.

4. Authors should validate the effect of 17 M1/M2 reprogramming compounds for their effect on changes on Arg and iNOS expression to provide strength to author's claim that these compounds can alter M1 state to M2 or vice-versa.

Response: Per reviewer's suggestion, we performed further validation by treating differentiated hMDM with 17 M1/M2 compounds followed by assaying selected M1 and M2 markers, including Arg and iNOS, by flow cytometry. We observed that iNOS was upregulated by 4 out of 17 M1-activating compounds but suppressed by 14 out of 17 M2-activating compounds. ARG1 was suppressed by 15 out of 17 M1-activating compounds and up-regulated by 3 out of 17 M2 compounds (Supplementary Fig. 5e). In addition, CD80 and CD86 were up-regulated by most of the M1-activating compounds and CD163 and CD206 were up-regulated by most of the M2-activating compounds, These results are included in the revised manuscript as Supplementary Fig. 5c-e.

5. Authors show in a co-culture of macrophages and B16F10 cells that thiostrepton activates and reprograms macrophages toward a pro-inflammatory state and enhances their tumoricidal activity in vitro. Author should perform this experiment in a quantitative setting and also look for definitive tumor cell killing, if at all, by thiostrepton treated macrophages in co-culture. This would be helpful in explaining results in fig.6.

Response: Fig. 5e showed that macrophages activated by thiostrepton could inhibit the tumor cell growth quantitatively in a dosage-dependent manner. Macrophages could kill tumor cells through cytokine-mediated cytotoxicity or antibody-dependent cellular phagocytosis (ADCP) or cytotoxicity (ADCC) when specific IgG exists to recognize tumor cells. We examined the effects of conditioned medium on tumor cell growth from macrophages either treated or untreated with thiostrepton. Mouse BMMs were treated with thiostrepton for 24 hrs (same as Fig. 5e), conditioned medium was collected and filtered. The same numbers of B16F10 melanoma cells were cultured for 12 hrs with conditioned medium or conditioned medium heated at 95°C for 5min. As expected, the medium from macrophages treated with thiostrepton inhibited the tumor cell growth compared to medium from untreated macrophages. But heat-inactivated conditioned medium did not inhibit the tumor cell growth (Supplementary Fig. 8a). These results suggest that thiostrepton treatment of macrophages induces the production of tumor growth inhibitor that is heat-labile.

To strengthen the conclusion with direct evidences, we performed ADCP with hMDM and mBMM. Briefly, tumor cells were labeled with eFluro670 and Ritumimab, then co-cultured for 2 hrs with hMDM or mBMM that have been primed with thiostrepton for 24 hrs. Macrophages with engulfed tumor cells (eFluro670+) among total macrophages (CD14+) were quantified by flow cytometry. Results showed that ~12% more thiostrepton-treated hMDM engulfed tumor cells than DMSO-treated hMDM (67.8% vs. 55.6%), while ~4% more thiostrepton-treated mBMM engulfed tumor cells than DMSO-

treated mBMM (30.3% vs. 26.5%). These results were included in the revised manuscript as Supplementary Fig. 8b,c.

6. While it is interesting to see decreased tumor growth, the exact state of the tumor-associated macrophages after treatment is not clear. These macrophages should be assessed with some activation markers in the flow cytometry data shown in Fig. 6d (like MHCII, CD206, CD86 mentioned in material and methods). This would allow identification whether thiostrepton is able to alter the activation state of TAMs in this mouse model.

Response: Per reviewer's suggestion, together with other reviewers' recommendations, we examined the expression of typical markers including intracellular Arg1, iNOS and surface CD86 in TAM from mice. Following treatment with thiostrepton alone or thiostrepton plus tumor-specific antibody (TA99), TAM expressed elevated levels of M1 markers iNOS and CD86 but a decreased level of M2 marker Arg1. These results suggest activation of TAM toward M1 state by thiostrepton. These results are included in the revised manuscript as Supplementary Fig. 10a-b.

7. Given the role of M1 macrophages in increasing adaptive immune response, which seems to be crucial in cancers with high mutation loads such as melanoma the adaptive immune response e.g., CD8 T cell frequencies and their activation in thiostrepton treated cohorts should be analyzed to strengthen these impressive in vivo results.

Response: Per reviewer's suggestion, we characterized tumor-infiltrated CD8+ T cells and NK cells. The frequencies of CD8+ T cells in the tumor did not have significant difference among different treatment groups, so as the proportions of CD8+ T cells that expressed TNF α and IFN γ . However, we observed significantly more TNF α and IFN γ expressing NK cells in the tumor in mice that were treated with thiostrepton alone or thiostrepton plus TA99. These results are included in the revised manuscript as Supplementary Fig. 10c.

8. Importantly, since thiostrepton could be acting on numerous cell types in vivo, a macrophage depletion strategy must be utilized to discern the contribution of macrophages in vivo to these results.

Response: We agree with the reviewer's concern. However, in B16F10 model, implanted tumors do not grow or grow very poorly if macrophages were depleted by clodronate liposome or anti-CSF-1R based on our own experience and previous publications (Banciu et al. 2008; Zhu et al. 2015, Moylihan et al. 2016). Therefore, it is difficult to assess the effect of thiostrepton if macrophages were depleted in B16F10 model. As an alternative, we used human B-cell lymphoma in NSG mice. Since NSG mice do not have T cells, B cells and NK cells, the effect of thiostrepton on lymphoma growth is unlikely through these immune cells. As shown in Supplementary Fig. 12,

thiostrepton inhibited lymphoma growth in the bone marrow, which is correlated with increased levels of macrophages and elevated expression of MHCII.

References:

Banciu et al. Antitumor Activity of Liposomal Prednisolone Phosphate Depends on the Presence of Functional Tumor-Associated Macrophages in Tumor Tissue. *Neoplasia*, 2008, 10(2):108.

Zhu et al. Synergistic Innate and Adaptive Immune Response to Combination Immunotherapy with Anti-Tumor Antigen Antibodies and Extended Serum Half-Life IL-2. *Cancer Cell*, 2015, 4:489.

Moylihan et al. Eradication of large established tumors in mice by combination immunotherapy that engages innate and adaptive immune responses. *Nature medicine*, 2016, 22:1402.

9. Also implies by the findings is relief of macrophage-mediated T cell suppression. Cd4 and CD8 depletion studies would need to be conducted to provide this data in support of the authors conjecture.

Response: We agree with the reviewer's comments. Macrophage activation toward M1 phenotype would relieve the macrophage-mediated suppression of T cells. Although we showed that there is no difference in the proportion of either infiltrated CD8+ T cells or TNF α +IFN γ + CD8+ T cells between thiostrepton versus vehicle group (Supplementary Fig. 10c), significantly more TNF α and IFN γ expressing NK cells were found in mice treated with thiostrepton alone or thiostrepton plus TA99. To further determine the role of T cells, it would be important to deplete T cells or combined with anti-PD1, but we think these studies are beyond the scope of the current study.

Minor revision:

1. The selection criteria for picking thiostrepton over for instance mocetinostat is not clear to the reviewer, as the latter seem to induce IFN γ signaling resembling recombinant IFN γ in vitro Fig. 2d. Authors should explain rationale for choosing thiostrepton, if any.

Response: We chose thiostrepton for the following reasons: First, thiostrepton upregulated several genes in macrophages with anti-tumor effects such as Tnf and Il1b, and induced smaller numbers of DEGs as compared to mocetinostat. Second, thiostrepton is an antibiotic and is known to inhibit both proteasome and FOXM1-associated DNA replication, which are very interesting targets for us. Besides thiostrepton, we tested several other compounds, including cucurbitacin I and mocetinostat, in in vivo studies. The inhibitory effect of cucurbitacin I on tumor growth in mice are shown in the originally submitted manuscript (Supplementary Fig. 13). As shown in the figure below, mocetinostat alone or in combination with TA99 also significantly inhibited tumor

growth in mice. Since mocetinostat is an HDAC inhibitor and induced a large number of differentially expressed genes, including both M1 and M2 genes (biased to M1) (Figure 2c-d and Supplementary Fig. 3a), we decided to focus on thiostrepton in the follow-up study. We have clarified the rationale for selecting thiostrepton for following up studies in the revised manuscript.

Figure R1. Tumor growth in B6 mice bearing subcutaneous B16F10 tumors treated i.p. with DMSO, TA99, mocetinostat (1mg/kg) and mocetinostat plus TA99 (n=5 mice per group). Arrows indicated the treatment time. * P<0.05, ** P<0.01. P values were calculated by t-test.

2. The authors use the term TAMs for mouse bone marrow macrophages exposed to IL4 and IL13, which would be incorrect as these are quintessential M2 as per also described at multiple points by authors in the manuscript. This need to be corrected to M2 macrophages, and referred to as bone marrow-derived macrophages (BMdM)

Response: In the revised manuscript, we refer IL-4/IL-13 polarized bone marrow-derived macrophages as M2, and tumor cell conditioned medium polarized bone marrow macrophages as TAM-like.

Reviewer #3 (M1/M2 macrophage, inflammation)(Remarks to the Author): The manuscript written by Dr. Hu et al. shows that they developed a technique to screen various chemicals and drugs' potentials to induce M1 or M2 macrophage phenotypes by analyzing changes of cell morphology. I feel this work is potentially interesting, however, there are some concerns.

Response: We thank the reviewer for the constructive comments and have revised the manuscript accordingly to address the following comments.

<Major remarks>

1) Authors certainly show that picked up compounds could change gene expressions of macrophages, however, in the current version there are few data to address whether the

molecules are changed in protein level, and where the point of action of the drug is. Regarding Bosutinib and Thiostrepton, authors should clarify the key molecules affected by each drug in the macrophage functional changes. By doing so, the reliability of the data will increase, which in turn would increase the reliability of this screening method.

Response: We have addressed the reviewer's concern in the following ways:

(1) Beside assaying changes at the transcriptional level following compound treatment as shown in Fig. 2, we have now validated the protein expression changes of some typical M1 markers (CD80, CD86) and M2 markers (CD206, CD163) by flow cytometry in hMDM treated with compounds. All six M1-activating compounds upregulated expression of CD80 and CD86 and down-regulated expression of CD163 and CD206. Both M2-activating compounds down-regulated CD80 and CD86 while one compound up-regulated CD163 (Supplementary Fig. 3c).

(2) Beside RNAseq in Fig. 4, we have performed further validations at both RNA level by qPCR and protein level by flow cytometry. At transcriptional level, M1 genes (CD86, TNF, IL1B and CXCL2) were up-regulated following treatment of hMDM with the selected 17 M1-activating compounds, whereas M2 genes (CD163, CD206 and IL10) were up-regulated by the selected 17 M2-activating compounds (Supplementary Fig. 5d). At the protein level, CD80 and CD86 were up-regulated by the 17 M1-activating compounds and down-regulated by the 17 M2-activating compounds, whereas CD163 and CD206 were up-regulated by the 17 M2-activating compounds and down-regulated by the 17 M1-activating compounds (Supplementary Fig. 5e).

(3) We also examined the effects of thiostrepton on the polarization of M2 macrophages that were generated by culturing with B16F10-derived tumor-conditioned medium. Arg1 protein level was reduced by thiostrepton while M1-activating markers MHCII, CD80 and iNOS were upregulated (Supplementary Fig. 7a).

(4) We also measured the M1 markers (iNOS and CD86) and M2 marker Arg1 in TAM from mice (Supplementary Fig. 10a-b).

(5) In addition, we have assayed changes at the protein level of selected M1 markers (CD80, CD86, HLA-DRB) and M2 markers (CD206 and CD163) following IFN- γ and IL-4 treatment of hMDM (Supplementary Fig. 1b).

All these results are included in the revised manuscript.

1-a) Is the tyrosine kinase that is the target of Bosutinib known generally as key molecules in macrophage polarization?

Response: Bosutinib inhibits multiple kinases, including Src kinases (Golas et al. 2003, Drugbank). Some of these tyrosine kinases, such as M-SCF receptor, BTK, SRC and MAPK, play critical role in macrophage polarization (Hamilton et al. 2014, Gabhann et al. 2014, Byeon et al. 2012, Zhang et al. 2019). A previous report suggested that bosutinib induces anti-inflammatory macrophages (M2) through targeting salt-induced

kinase (SIK) (Ozanne et al. 2015), which belongs to AMPK family. Our data suggests that bosutinib regulates cell activation and protein secretion pathways, both of which are associated with macrophage activation. These results suggest that bosutinib may regulate macrophage activation and programming by inhibiting multiple kinases.

Reference:

Golas et al. SKI-606, a 4-anilino-3-quinolinecarbonitrile dual inhibitor of Src and Abl kinases, is a potent antiproliferative agent against chronic myelogenous leukemia cells in culture and causes regression of K562 xenografts in nude mice. *Cancer Research*, 2003, 63(2):375.

Drugbank: <https://www.drugbank.ca/drugs/DB06616>

Hamilton et al. Myeloid colony-stimulating factors as regulators of macrophage polarization. *Frontiers in immunology*, 2014, 5:554

Gabhann et al. Btk regulates macrophage polarization in response to lipopolysaccharide. *PloS one*, 2014 9: e85834.

Byeon et al. The Role of Src Kinase in Macrophage-Mediated Inflammatory Responses. 2012, 512926

Zhang et al. Targeting MAPK Pathways by Naringenin Modulates Microglia M1/M2 Polarization in Lipopolysaccharide-Stimulated Cultures. *Frontiers in Cellular Neuroscience*, 2019, 12:531

Ozanne et al. The clinically approved drugs dasatinib and bosutinib induce anti-inflammatory macrophages by inhibiting the salt-inducible kinases. *The Biochemical journal*, 2015, 465: 271.

1-b) If so, please show that Bosutinib effect on the activation of the kinase (ex. phosphorylation), and the inhibitor of the kinase can cancel the effect of Bosutinib and M2 polarization.

Response: Following their initial publication (Ozanne et al. 2015 *Biochemistry J.*), Clark and colleagues further investigated how SIK regulates macrophage polarization in 2017 (Darling et al. 2017). They showed that SIK-deficient macrophages exhibit M2-polarizing phenotype. Consistently, bosutinib induced the dephosphorylation of CRTC3 which is required for its nuclear translocation by dissociating from protein 14-3-3 to elevate the expression of anti-inflammatory IL-10. Before this publication, we also performed some preliminary studies and our data showed (Figure R2): 1) in human monocyte cell line (THP1), bosutinib induced dephosphorylation of SRC following 3 or 24 hrs treatment; 2) bosutinib induced phosphorylation of SIK1/2/3 in 3 or 24 hrs; 3) NaCl suppressed the phosphorylation of SIK, 3) salt treatment promotes macrophage to acquire pro-inflammation phenotype (Figure R2).

Figure R2. **a**, Western blotting of phosphorylation of SRC and SIKs in THP1 cells treated with bosutinib. THP1 cells were treated with DMSO (control), bosutinib (1 μ M) or NaCl (200 mM) or bosutinib plus NaCl for 3 or 24 hrs. Cell lysates were subjected to Western blotting with the indicated antibodies. **b**, Gene set enrichment analysis of NaCl-induced gene expression in hMDM. hMDM were cultured in complete RPMI medium or the same medium plus 200 mM NaCl for 24 hrs. RNAseq was performed and compared between the two samples.

Reference:

Darling et al. Inhibition of SIK2 and SIK3 during differentiation enhances the anti-inflammatory phenotype of macrophages. *Biochem journal*, 2017, 474:521.

1-c) As the same concept as 1-a) and 1-b), what molecule is the target of Thiostrepton? Is the molecule essential for M1 polarization?

Response: Thiostrepton have been reported to binds to proteasome and/or FOXM1 (Ref. 30-32). Consistent with these findings, we show thiostrepton upregulated proteasome pathway and suppressed FOXM1 pathway in macrophages (Fig. 5). Proteasome is essential for proinflammatory pathways mediated by TLR/NF κ B (Quershi et al. 2012). However, it is unknown whether and how FOXM1 or DNA replication regulates macrophage activation/polarization.

Reference:

Quershi et al. Proteasome protease mediated regulation of cytokine induction and inflammation. 2012, *Biochimica et Biophysica Acta* 1823:2087

2) In the *in vivo* tumor model in Fig. 6, Thio shows the antitumor effect, and it can also activate M1 macrophages. With these two data, the authors state that Thio induces M1 macrophages *in vivo*, which is effective against the tumor. However, as the authors discussed, there is a possibility that Thio suppresses tumor growth regardless of macrophage activation. To verify this, I recommend additional experiments.
 2-a) Thio-primed macrophages are injected into the tumor and see the anti-tumor effects.

Response: Several clinic trials involving adoptively transferring macrophages polarized with proinflammatory cytokines in vitro were conducted in the 1990s. None of the trials showed any significant anti-tumor effect, because macrophages quickly reverted to M2 phenotypes once trafficked into the tumor microenvironment (see review Lee et al. 2016). Based on these previous studies, adoptively transfer of thiostrepton-primed macrophages may not work to confirm its effect in vivo.

Reference:

Lee et al. Macrophage-Based Cell Therapies: The Long and Winding Road. *Journal of Control Release*, 2016, 240:527.

2-b) Macrophages are depleted by liposomes or other reagents, then does the anti-tumor effect of Thio disappear?

Response: As we responded to reviewer 2. Unfortunately, in B16F10 model, implanted tumors do not grow or grow very poorly if macrophages were depleted by clodronate liposome or anti-CSF-1R based on our own experience and previous publications (Banciu et al. 2008; Zhu et al. 2015, Moylihan et al. 2016). Therefore, it is difficult to assess the effect of thiostrepton if macrophages were depleted in B16F10 model. As an alternative, we used human B-cell lymphoma in NSG mice. Since NSG mice do not have T cells, B cells and NK cells, the effect of thiostrepton on lymphoma growth is unlikely through these immune cells. As shown in Supplementary Fig. 12, thiostrepton inhibited lymphoma growth in the bone marrow, which is correlated with increased levels of macrophages and elevated expression of MHCII.

References:

Banciu et al. Antitumor Activity of Liposomal Prednisolone Phosphate Depends on the Presence of Functional Tumor-Associated Macrophages in Tumor Tissue. *Neoplasia*, 2008, 10(2):108.

Zhu et al. Synergistic Innate and Adaptive Immune Response to Combination Immunotherapy with Anti-Tumor Antigen Antibodies and Extended Serum Half-Life IL-2. *Cancer Cell*, 2015, 4:489.

Moylihan et al. Eradication of large established tumors in mice by combination immunotherapy that engages innate and adaptive immune responses. *Nature medicine*, 2016, 22:1402.

<Minor remarks>

In Fig 1a and 1b, Z-score of M1 cells is negative (Z=-xx). On the other hand, in Fig 1e, Z-score of Compound 1, which can induce M0 to M1, is positive (Z=12). I have read the figure legend, Fig 1e shows the difference in cell morphologies comparing to DMSO control, but I could not understand why the Z score of compound 1 move positive compared with DMSO. I guess I've misunderstood something, please expand results or legends so that the reader can easily understand them.

Response: We are sorry that the two figures were put in the wrong order. We have corrected this error in the revised manuscript.

REVIEWERS' COMMENTS

Reviewer #1 (Remarks to the Author):

The authors have addressed concerns raised.

Reviewer #3 (Remarks to the Author):

I have read the revised manuscript and comments of response written by Dr. Hu et al. I am satisfied with the authors' response. Authors provided literature evidence and additional experiments on the effects of the compounds on macrophages. They also provided rationales for the difficulty of conducting additional experiments in mice.